# Lotka-Volterra pairwise modeling fails to capture diverse pairwise microbial interactions

**Babak Momeni[1,2]\*, Li Xie[2], Wenying Shou[2]\***

[1]Department of Biology, Boston College, Chestnut Hill, United States; [2]Division of Basic Sciences, Fred Hutchinson Cancer Research Center, Seattle, United States

**Abstract** Pairwise models are commonly used to describe many-species communities. In these models, an individual receives additive fitness effects from pairwise interactions with each species in the community ('additivity assumption'). All pairwise interactions are typically represented by a single equation where parameters reflect signs and strengths of fitness effects ('universality assumption'). Here, we show that a single equation fails to qualitatively capture diverse pairwise microbial interactions. We build mechanistic reference models for two microbial species engaging in commonly-found chemical-mediated interactions, and attempt to derive pairwise models. Different equations are appropriate depending on whether a mediator is consumable or reusable, whether an interaction is mediated by one or more mediators, and sometimes even on quantitative details of the community (e.g. relative fitness of the two species, initial conditions). Our results, combined with potential violation of the additivity assumption in many-species communities, suggest that pairwise modeling will often fail to predict microbial dynamics.

**\*For correspondence:** momeni@bc.edu (BM); wenying.shou@gmail.com (WS)

## Introduction

Multispecies microbial communities are ubiquitous. Microbial communities are important for industrial applications such as cheese and wine fermentation (*van Hijum et al., 2013*) and municipal waste treatment (*Seghezzo et al., 1998*). Microbial communities are also important for human health: they can modulate immune responses and food digestion (*Round and Mazmanian, 2009*; *Kau et al., 2011*) during health and disease. Properties of the entire community ('community properties', e.g. species dynamics, ability to survive internal or external perturbations, and biochemical activities of the entire community) are influenced by interactions wherein individuals alter the physiology of other individuals (*Widder et al., 2016*). To understand and predict community properties, choosing the appropriate mathematical model to describe species interactions is critical.

A mathematical model ideally focuses only on details that are essential to community properties of interest. However, it is often unclear *a priori* what the minimal essential details are. We define 'mechanistic models' as models that explicitly consider interaction mediators as state variables. For example, if species $S_1$ releases a compound $C_1$ which stimulates species $S_2$ growth upon consumption by $S_2$, then a mechanistic model tracks concentrations of $S_1$, $C_1$, and $S_2$ (*Figure 1A and B*, left panels). Note that mechanistic models used here still omit molecular details such as how chemical mediators are received and processed by recipients and how mediators subsequently act on recipients. In contrast, Lotka-Volterra ('L-V') pairwise models only consider the fitness effects of interactions. Specifically, L-V models assume that the fitness of an individual is the sum of its basal fitness (the net growth rate of an individual in isolation) and fitness influences from pairwise interactions with individuals of the same species and of every other species in the community ('additivity' assumption). Furthermore, regardless of interaction mechanisms or quantitative details of a

**eLife digest** From the soil to our body, microbes, such as bacteria, are everywhere and affect us in many ways. Many microbes perform important roles in natural environments and for our health, but some of them can cause harm and lead to diseases. Often, microbes affect and interact with each other within large groups or communities. Because of their widespread ramifications, it is important to understand how microbial communities work.

In addition to experiments, mathematical modeling offers one way to gain insight into the dynamics of microbial communities. A model commonly used to describe the interactions between organisms is the so-called 'pairwise model'. Pairwise models can be useful to predict the dynamics of a community in which two species physically interact, such as a predator-prey community. However, it was unknown if this model was suitable to adequately predict the dynamics of microbial species in communities. Microbes often interact via chemicals that diffuse in the environment. For example, one microbe might provide food for another microbe or release toxins to kill it. However, a pairwise model does not consider food or toxins, but only how one microbe stimulates or inhibits the growth of another.

Momeni et al. simulated different scenarios commonly found in microbial communities to test whether a pairwise model could capture how, for example, chemicals released by one bacterial species would either help others to grow or stop them from growing. The results showed that for many scenarios, pairwise models cannot qualitatively represent the dynamics of a microbial community.

A next step will be to work on the limitations of current experimental technologies and mathematical models to improve the understanding of microbial communities. This knowledge could be used to develop new strategies for ecosystem engineering, such as for example making soils more fertile to improve crop yields, or tackling antibiotic resistance of bacteria.

community, all fitness influences are typically expressed using a single equation form wherein parameters can vary to reflect the signs and magnitudes of fitness influences ('universality' assumption). Thus in the example above, a pairwise model only describes how $S_1$ increases the fitness of $S_2$ (*Figure 1A and B*, right panels).

L-V pairwise models are popular. L-V pairwise modeling has successfully explained the oscillatory dynamics of hare and its predator lynx (*Figure 1—figure supplement 1*) (*Volterra, 1926*; *Wangersky, 1978*; *BiologyEOC, 2016*). Pairwise models have also been instrumental in delineating conditions for multiple carnivores to coexist when competing for herbivores (*MacArthur, 1970*; *Chesson, 1990*). In both cases, mechanistic models and pairwise models happen to be mathematically equivalent for the following reasons. In the hare-lynx example, both species are also interaction mediators, and therefore pairwise and mechanistic models are identical. In the second example, if herbivores (mediators of competitive interactions between carnivores) rapidly reach steady state, herbivores can be mathematically eliminated from the mechanistic model to yield a pairwise model of competing carnivores (*MacArthur, 1970*; *Chesson, 1990*). Pairwise models are often used to predict how perturbations to steady-state species composition exacerbate or decline over time (*May, 1972*; *Thébault and Fontaine, 2010*; *Mougi and Kondoh, 2012*; *Allesina and Tang, 2012*; *Suweis et al., 2013*; *Coyte et al., 2015*). Although most work are motivated by contact-dependent prey-predation (e.g. hare-lynx) or mutualisms (e.g. plant-pollinator) where L-V models could be identical to mechanistic models, these work do not explicitly exclude chemical-mediated interactions where species are distinct from interaction mediators.

The temptation of using pairwise models is indeed high, including in microbial communities where many interactions are mediated by chemicals (*Mounier et al., 2008*; *Faust and Raes, 2012*; *Stein et al., 2013*; *Marino et al., 2014*; *Coyte et al., 2015*). Even though pairwise models do not capture the dynamics of chemical mediators, predicting species dynamics is still highly desirable in, for example, forecasting species diversity and compositional stability. For chemical-mediated interactions, L-V pairwise models are far easier to construct than mechanistic models for the following reasons. Mechanistic models would require knowledge of chemical mediators, which are often

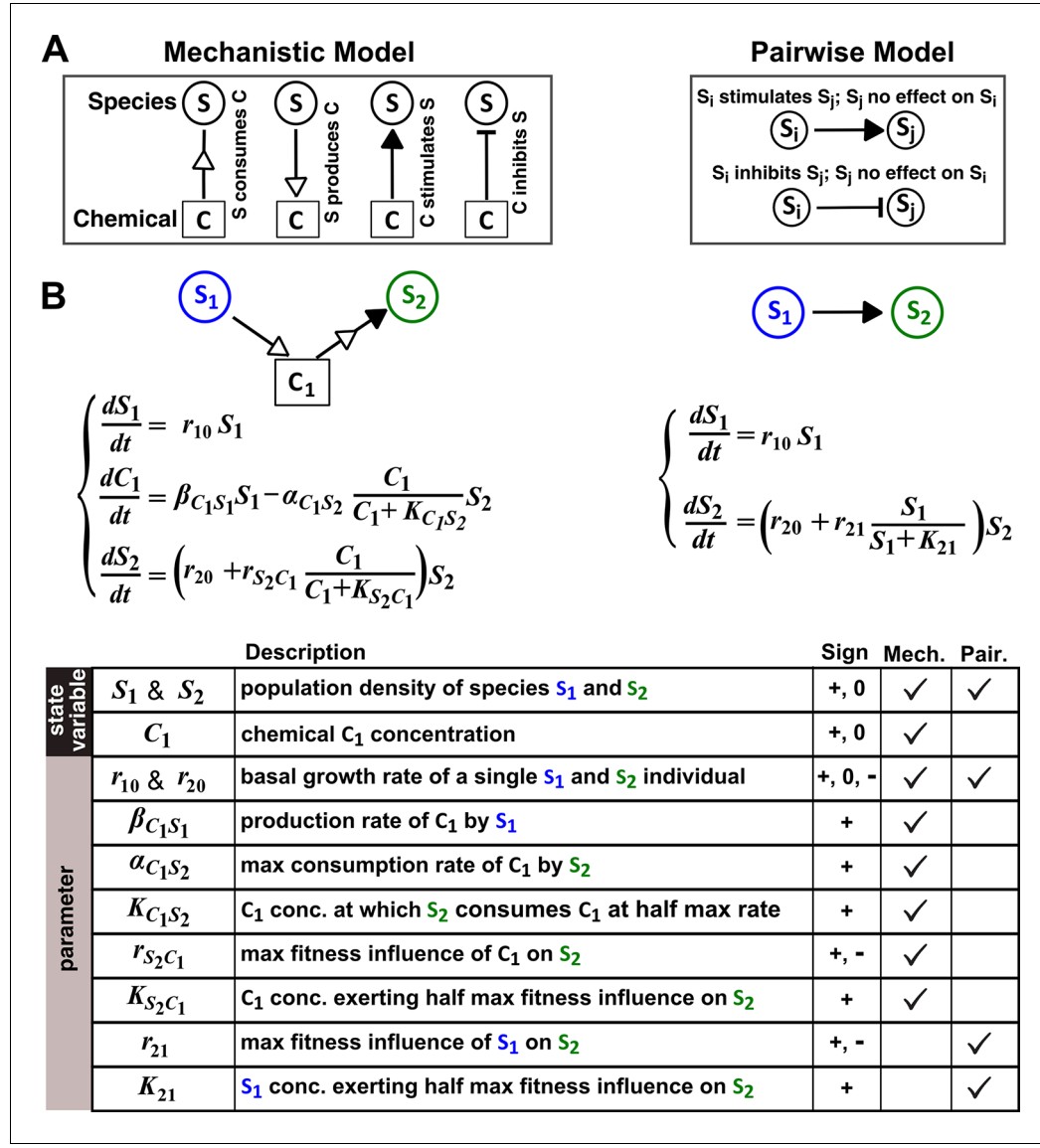

**Figure 1.** The abstraction of interaction mechanisms in a pairwise model compared to a mechanistic model. (A) The mechanistic model (left) considers a bipartite network of species and chemical interaction mediators. A species can produce or consume chemicals (open arrowheads pointing towards and away from the chemical, respectively). A chemical mediator can positively or negatively influence the fitness of its target species (filled arrowhead and bar, respectively). The corresponding L-V pairwise model (right) includes only the fitness effects of species interactions, which can be positive (filled arrowhead), negative (bar), or zero (line terminus). (B) In the example here, species $S_1$ releases chemical $C_1$, and $C_1$ is consumed by species $S_2$ and promotes $S_2$'s fitness. In the mechanistic model, the three equations respectively state that (1) $S_1$ grows exponentially at a rate $r_{10}$, (2) $C_1$ is released by $S_1$ at a rate $\beta_{C_1S_1}$ and consumed by $S_2$ with saturable kinetics (maximal consumption rate $\alpha_{C_1S_2}$ and half-saturation constant $K_{C_1S_2}$), and (3) $S_2$'s growth (basal fitness $r_{20}$) is influenced by $C_1$ in a saturable fashion. In the pairwise model here, the first equation is identical to that of the mechanistic model. The second equation is similar to the last equation of the mechanistic model except that $r_{21}$ and $K_{21}$ together reflect how the density of $S_1$ ($S_1$) affects the fitness of $S_2$ in a saturable fashion. For all parameters with double subscripts, the first subscript denotes the focal species or chemical, and the second subscript denotes the influencer. Note that unlike in mechanistic models, we have omitted '$S$' from subscripts in pairwise models (e.g. $r_{21}$ instead of $r_{S_2S_1}$) for simplicity. In this example, both $r_{21}$ and $r_{S_2S_1}$ are positive.

The following figure supplements are available for figure 1:

*Figure 1 continued on next page*

*Figure 1 continued*

**Figure supplement 1.** An L-V pairwise model successfully predicts oscillations in population dynamics of the hare-lynx prey-predator community.
**Figure supplement 2.** Deriving a pairwise model.

challenging to identify. Since chemical mediators are explicitly modeled, mechanistic models require more equations and parameters than their cognate pairwise models (*Figure 1*, Table). Pairwise model parameters are relatively easy to estimate using community dynamics or dynamics of monocultures and pairwise cocultures (*Mounier et al., 2008*; *Stein et al., 2013*; *Guo and Boedicker, 2016*). Consequently, pairwise modeling has been liberally applied to microbial communities.

L-V pairwise models have been criticized when applied to communities of more than two species (referred to as '*multispecies communities*') (*Levine, 1976*; *Tilman, 1987*; *Wootton, 1993*, *2002*; *Werner and Peacor, 2003*; *Stanton, 2003*; *Schmitz et al., 2004*). This is because a third species can influence interactions between a species pair ('indirect interactions'), which sometimes violates the additivity assumption of pairwise models. For example, a carnivore can indirectly increase the density of a plant by decreasing the density of an herbivore ('interaction chain'; 'density-mediated indirect interactions'). A carnivore can also decrease how often an herbivore forages plants ('interaction modification', 'trait-mediated indirect interactions', or 'higher order interactions') (*Vandermeer, 1969*; *Wootton, 1994*; *Billick and Case, 1994*; *Wootton, 2002*). In interaction modification, foraging per herbivore decreases, whereas in interaction chain, the density of herbivores decreases. Interaction modification (but not interaction chain) violates the additivity assumption (Methods-Interaction modification but not interaction chain violates the additivity assumption) (*Tilman, 1987*; *Wootton, 1994*; *Schmitz et al., 2004*) and can cause the pairwise model to generate qualitatively wrong predictions. Indeed, pairwise models largely failed to predict biomass and species coexistence in three-species and seven-species plant communities (*Dormann and Roxburgh, 2005*), although reported failures of pairwise models could be due to limitations in data collection and analysis (*Case and Bender, 1981*; *Billick and Case, 1994*).

Here, we examine the universality assumption of pairwise models when applied to microbial communities (or any community that employs diverse chemical-mediated interactions). Microbes often influence other microbes in a myriad of fashions, via consumable metabolites, reusable signaling molecules, or a combination of chemicals (*Figure 2*). *Can a single equation form, traditionally employed in pairwise models, qualitatively describe diverse interactions between two microbial species*? The answer is unclear. On the one hand, pairwise models have been applied successfully to diverse microbial communities. For example, an L-V pairwise model and a mechanistic model both correctly predicted ratio stabilization and spatial intermixing between two strongly-cooperating populations exchanging diffusible essential metabolites (*Momeni et al., 2013*). In other examples, pairwise models largely captured competition outcomes and metabolic activities of three-species and four-species artificial microbial communities (*Vandermeer, 1969*; *Guo and Boedicker, 2016*; *Friedman et al., 2017*). On the other hand, pairwise models often failed to predict species coexistence in seven-species microbial communities (*Friedman et al., 2017*), although this could be due to interaction modification discussed above.

Instead of investigating natural communities where interaction mechanisms can be difficult to identify, we use *in silico* communities. In these communities, two species interact via mechanisms commonly encountered in microbial communities, including growth-promoting and growth-inhibiting interactions mediated by reusable and consumable compounds (*Figure 2*) (*Stams, 1994*; *Czárán et al., 2002*; *Duan et al., 2009*). We construct mechanistic models for these two-species communities and attempt to derive from them pairwise models. A mechanistic reference model offers several advantages: community dynamics is deterministically known; deriving a pairwise model is not limited by inaccuracy of experimental methods; and the flexibility in creating different reference models allows us to explore a variety of interaction mechanisms. We demonstrate that a single pairwise equation form often fails for commonly-encountered diverse pairwise microbial interactions. We conclude by discussing when pairwise models might or might not be useful, in light of our findings.

**Figure 2.** Chemical-mediated interactions commonly found in microbial communities. Interactions can be intra- or inter-population. Examples are meant to be illustrative instead of comprehensive.

## Results

Throughout this work, we consider communities grown in a well-mixed environment where all individuals interact with each other with an equal chance. A well-mixed environment can be found in industrial fermenters. Moreover, at a sufficiently small spatial scale, a spatially-structured environment can be approximated as a well-mixed environment, as chemicals are uniformly-distributed locally. Motile organisms also reduce the degree of spatial structure. A well-mixed environment allows us to use ordinary differential equations (ODEs), which are more tractable than partial differential equations demanded by a spatially-structured environment. This in turn allows us to sometimes analytically demonstrate failures of pairwise models.

### Mechanistic model versus pairwise model

A mechanistic model describes how species release or consume chemicals and how chemicals stimulate or inhibit species growth (*Figure 1A* left). In contrast, in pairwise models, interaction mediators are not explicitly considered (*Figure 1A* right). Instead, the growth rate of an individual of species $S_i$ is the sum of its basal fitness ($r_{i0}$, net growth rate of the individual in the absence of any intra-species or inter-species interactions) and fitness effects from intra-species and inter-species interactions. The fitness effect from species $S_j$ to species $S_i$ is represented by $f_{ij}(S_j)$, where $S_j$ is the density of species $S_j$. $f_{ij}(S_j)$ is a linear or nonlinear function of only $S_j$ and not of another species. When $j = i$, $f_{ii}(S_i)$ represents density-dependent fitness effect within $S_i$ (e.g. density-dependent growth inhibition or stimulation).

In a multi-species pairwise model, a single form of $f_{ij}$ is used for all pairwise species interactions. For example, the most popular L-V model is linear L-V:

$$\frac{dS_i}{dt} = \left[ r_{i0} + \sum_j r_{ij} S_j \right] S_i \qquad (1)$$

Here, $r_{i0}$ is the basal fitness of an individual of $S_i$, and can be positive, negative, or zero; $r_{ij}$ is the fitness effect per $S_j$ individual on $S_i$. Positive, negative, or zero $r_{ij}$ represents growth stimulation,

inhibition, or no effect, respectively. An example of linear L-V is the logistic L-V pairwise model tradi-
tionally used for competitive communities:

$$\frac{dS_i}{dt} = r_{i0}\left[1 - \sum_j \frac{S_j}{\Lambda_{ij}}\right]S_i \tag{2}$$

Here, nonnegative $r_{i0}$ is the basal fitness of $S_i$; positive $\Lambda_{ij}$ is the carrying capacity imposed by lim-
iting shared resource (e.g. space or carbon source) such that a single $S_i$ individual will have a zero
net growth rate when competing with a total of $\Lambda_{ij}$ individuals of $S_j$.

Alternative forms of fitness effect $f_{ij}$ (*Wangersky, 1978*) include L-V with delayed influence, where
the fitness influence of one species on another may lag in time (*Gopalsamy, 1992*), or saturable L-V
(*Thébault and Fontaine, 2010*) where

$$\frac{dS_i}{dt} = \left[r_{i0} + \sum_j r_{ij}\frac{S_j}{K_{ij}+S_j}\right]S_i \tag{3}$$

Here, $r_{i0}$ is the basal fitness of an individual of $S_i$, $r_{ij}$ is the maximal fitness effect species $S_j$ can
exert on $S_i$, and $K_{ij}$ (>0) is the $S_j$ at which half maximal fitness effect on $S_i$ is achieved. $r_{i0}$ and $r_{ij}$ can
be positive, negative, or zero. Note that at a low concentration of influencer, the saturable form can
be converted to a linear form.

Our goal is to test whether a single equation form of pairwise model can qualitatively predict
dynamics of species pairs engaging in various types of interactions commonly found in microbial
communities (e.g. *Figure 2*). To do so, we use a combination of analytical and numerical approaches
(*Figure 1—figure supplement 2*). Analytically deriving a pairwise model from a mechanistic model
not only reveals assumptions required to generate the pairwise model, but also alleviates any con-
cern that we may have failed to identify the optimal pairwise model parameters. When interactions
become more complex (e.g. involving multiple mediators), we take the numerical approach, which is
typically used to infer pairwise models from experimental results (*Pascual and Kareiva, 1996*). In the
numerical approach, we mimic experimentalists by first deciding on a pairwise model to be used,
and then employing a nonlinear least squares routine to numerically identify model parameters that
minimize the average difference $\bar{D}$ between pairwise and mechanistic model dynamics within a train-
ing time window $T$ (*Figure 1—figure supplement 2*; Methods-Summary of simulation files). To eval-
uate how well a pairwise model predicts long-term mechanistic model dynamics, we 'buy time' by
introducing 'dilutions' in numerical simulations of both models and quantify their difference $\bar{D}$.

## Reusable versus consumable mediators require different pairwise models

In this section, we analytically derive pairwise models from mechanistic models of two-species com-
munities where one species affects the other species through a single mediator. The mediator is
either reusable such as signaling molecules in quorum sensing (*Duan et al., 2009*; *Jakubovics, 2010*)
or consumable such as metabolites (*Stams, 1994*; *Freilich et al., 2011*) (*Figure 2*). We show that a
single pairwise model may not encompass these different interaction mechanisms and that for con-
sumable mediator, the choice of pairwise model also depends on details such as the relative fitness
and initial densities of the two species.

Consider a commensal community where species $S_1$ stimulates the growth of species $S_2$ by pro-
ducing a reusable (*Figure 3A*) or a consumable (*Figure 3B*) chemical $C_1$. We consider community
dynamics where species are not limited by any abiotic resources, such as within a dilution cycle of a
turbidostat experiment where all other metabolites are in excess.

When $C_1$ is reusable, the mechanistic model (*Figure 3A,i*) can be transformed into a saturable L-V
pairwise model (compare *Figure 3A,ii* with *Equation 3*), especially after the concentration of the
mediator (which is initially zero) has acclimated to be proportional to the producer population size
(*Figure 3A* legend; *Figure 3—figure supplement 1*). This saturable L-V pairwise model is valid
regardless of whether the producer coexists with the consumer, outcompetes the consumer, or is
outcompeted by the consumer.

If $C_1$ is consumable, different scenarios are possible (*Figure 3B*; Methods).

Case I: When supplier $S_1$ always grows faster than consumer $S_2$ (the basal fitness of $S_1$ is higher
than the maximal fitness of $S_2$), $C_1$ will eventually accumulate proportionally to $S_1$ (*Figure 4A* left;

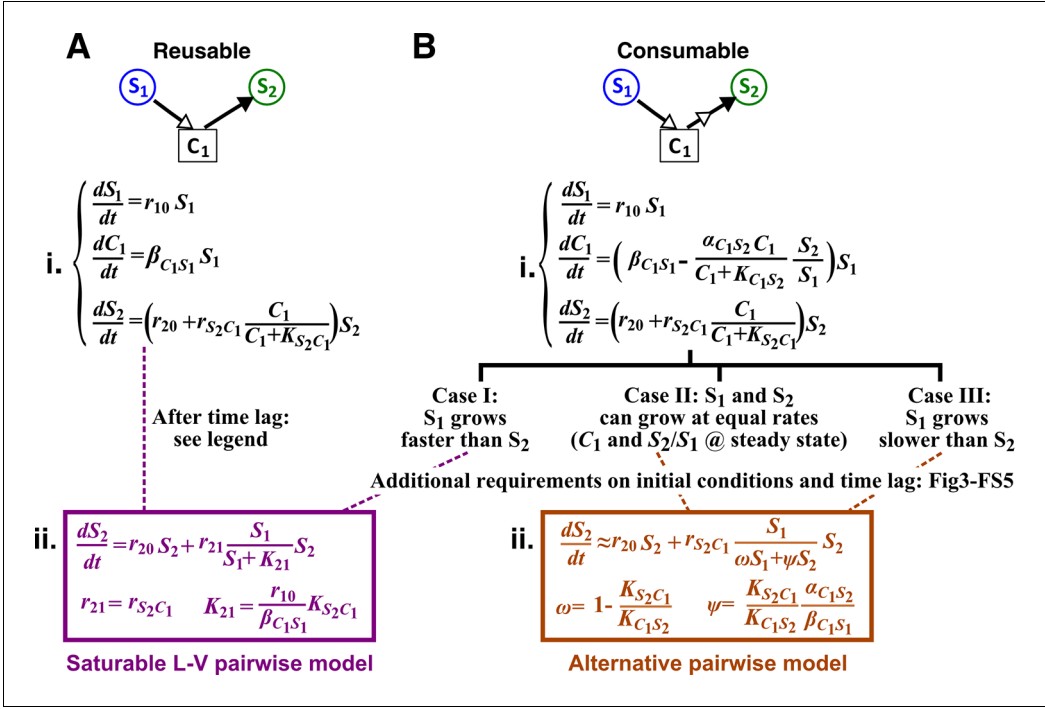

**Figure 3.** Interactions mediated via a single mediator are best represented by different forms of pairwise models, depending on whether the mediator is consumable or reusable and on the relative fitness and initial densities of the two species. $S_1$ stimulates the growth of $S_2$ via a reusable (**A**) or a consumable (**B**) chemical $C_1$. In mechanistic models of the two cases (i), equations for $S_1$ and $S_2$ are identical but equations for $C_1$ are different. In (**A**), $C_1$ can be solved to yield $C_1 = (\beta_{C_1S_1}/r_{10})S_{10}\exp(r_{10}t) - (\beta_{C_1S_1}/r_{10})S_{10} = (\beta_{C_1S_1}/r_{10})S_1 - (\beta_{C_1S_1}/r_{10})S_{10}$, assuming zero initial $C_1$. Here, $S_{10}$ is $S_1$ at time zero. We have approximated $C_1$ by omitting the second term (valid after the initial transient response has passed so that $C_1$ has become proportional to $S_1$). This approximation allows an exact match between the mechanistic model and the saturable L-V pairwise model (**ii**). In (**B**), depending on the relative growth rates of the two species, and if additional requirements are satisfied (Methods; *Figure 3—figure supplement 2*; *Figure 3—figure supplement 3*; *Figure 3—figure supplement 4*; *Figure 3—figure supplement 5*), either saturable L-V or alternative pairwise model should be used.

The following source data and figure supplements are available for figure 3:

**Source data 1.** List of parameters for simulations in *Figure 3—figure supplement 1*.

**Source data 2.** List of parameters for simulations in *Figure 3—figure supplement 2* on interactions through a consumable mediator.

**Source data 3.** List of parameters for simulations in *Figure 3—figure supplement 3* on conditions required for convergence of the alternative pairwise model.

**Source data 4.** List of parameters for simulations in *Figure 3—figure supplement 4* on how dilution might affect the convergence of a pairwise model.

**Figure supplement 1.** For a reusable mediator, parameter estimation after acclimation time leads to a more accurate saturable L-V pairwise model.

**Figure supplement 2.** Community trajectory approaching the -zero-isocline allows us to use the alternative pairwise model approximation.

**Figure supplement 3.** Condition for the alternative pairwise model to converge to the mechanistic model in the absence of dilutions.

*Figure 3 continued on next page*

*Figure 3 continued*

**Figure supplement 4.** Initial conditions that require long convergence time and thus dilutions may prevent the alternative pairwise model to converge to the mechanistic model.

**Figure supplement 5.** Additional requirements for deriving a pairwise model from a mechanistic model, when $S_1$ affects $S_2$ via a single consumable mediator $C_1$ where $C_1(0) = 0$.

Methods-Deriving a pairwise model for interactions mediated by a single consumable mediator Case I). In this case, $C_1$ may be approximated as a reusable mediator and can be predicted by the saturable L-V pairwise model (*Figure 4A* right, compare dotted and solid lines).

Case II: When $S_1$ and $S_2$ can coexist (the basal fitness of $S_1$ is higher than the basal fitness of $S_2$ but less than the maximal fitness of $S_2$), a steady state solution for $C_1$ and species ratio $R_S = S_2/S_1$ exists (*Figure 4B*; Methods-Deriving a pairwise model for interactions mediated by a single consumable mediator Case II, *Equation 11*). To arrive at a pairwise model, we will need to eliminate $C_1$ which is mathematically possible (i.e. after community dynamics converges to the '$f$-zero-isocline' on the phase plane of mediator $C_1$ and species ratio $R_S$, as depicted by blue lines in *Figure 3—figure supplement 2A–D*). However, the derived pairwise model differs from the saturable L-V model:

$$\frac{dS_2}{dt} = r_{20}S_2 + \frac{r_{S_2 C_1} S_1}{\omega S_1 + \psi S_2} S_2 \tag{4}$$

where constants $r_{20}$, $r_{S_2 C_1}$, and $\omega = 1 - K_{S_2 C_1}/K_{C_1 S_2}$ can be positive, negative, or zero, and $\psi = (K_{S_2 C_1} \alpha_{C_1 S_2})/(K_{C_1 S_2} \beta_{C_1 S_1})$ is positive (see *Figure 1* table for parameter definitions and see *Equation 13* in Methods). We will refer to this equation as 'alternative pairwise model', although the fitness influence term is a function of both $S_1$ and $S_2$ instead of the influencer $S_1$ alone as defined in the traditional L-V pairwise model.

Case III: When supplier $S_1$ always grows slower than consumer $S_2$, i.e. when the basal fitness of $S_1$ ($r_{10}$) is less than the basal fitness of $S_2$ ($r_{20}$), consumable $C_1$ declines to zero concentration. This is because $C_1$ is consumed by $S_2$ whose relative abundance over $S_1$ eventually exponentially increases at a rate of $r_{20} - r_{10}$. Similar to Case II, under certain conditions (i.e. after community dynamics converges to the $f$-zero-isocline as seen in *Figure 3—figure supplement 2E–H*), the alternative pairwise model (*Equation 4*) can be derived (Methods-Deriving a pairwise model for interactions mediated by a single consumable mediator, Case III).

For both Case II and Case III, we analytically demonstrate that in the absence of dilutions, alternative pairwise model dynamics can converge to mechanistic model dynamics (see *Figure 3—figure supplements 3* and *5* for initial condition requirement and time scale of convergence). However, if initial $S_1$ and $S_2$ are such that the time scale of convergence is long compared to the duration of one dilution cycle (e.g. *Figure 3—figure supplement 2C and G*), then we will have to perform dilutions and the saturable L-V model can sometimes be more appropriate than the alternative model (*Figure 3—figure supplement 4*). Thus in these cases, whether a saturable L-V or an alternative model is more appropriate also depends on initial conditions.

The alternative model (*Equation 4*) can be further simplified to

$$dS_2/dt = (r_{20} + \rho S_1/S_2)S_2 \tag{5}$$

if additionally, the half-saturation constant $K$ for $C_1$'s consumption ($K_{C_1 S_2}$) is identical to that for $C_1$'s influence on the growth of consumer ($K_{S_2 C_1}$), and if $S_2$ has not gone extinct. This equation form has precedence in the literature (e.g. [*Mougi and Kondoh, 2012*]), where the interaction strength $r_{21}$ reflects the fact that the consumable mediator from $S_1$ is divided among consumer $S_2$. Thus, we can regard the alternative model (*Equation 4*) or its simplified version (*Equation 5*) as a 'divided influence' model.

The saturable L-V model and the alternative model are not interchangeable (*Figure 4*). When a consumable mediator accumulates without reaching a steady state within each dilution cycle (*Figure 4A* left; inset: $C_1$ eventually becomes proportional to $S_1$), the saturable L-V model is predictive of community dynamics (*Figure 4A* right, compare dotted and solid lines). In contrast,

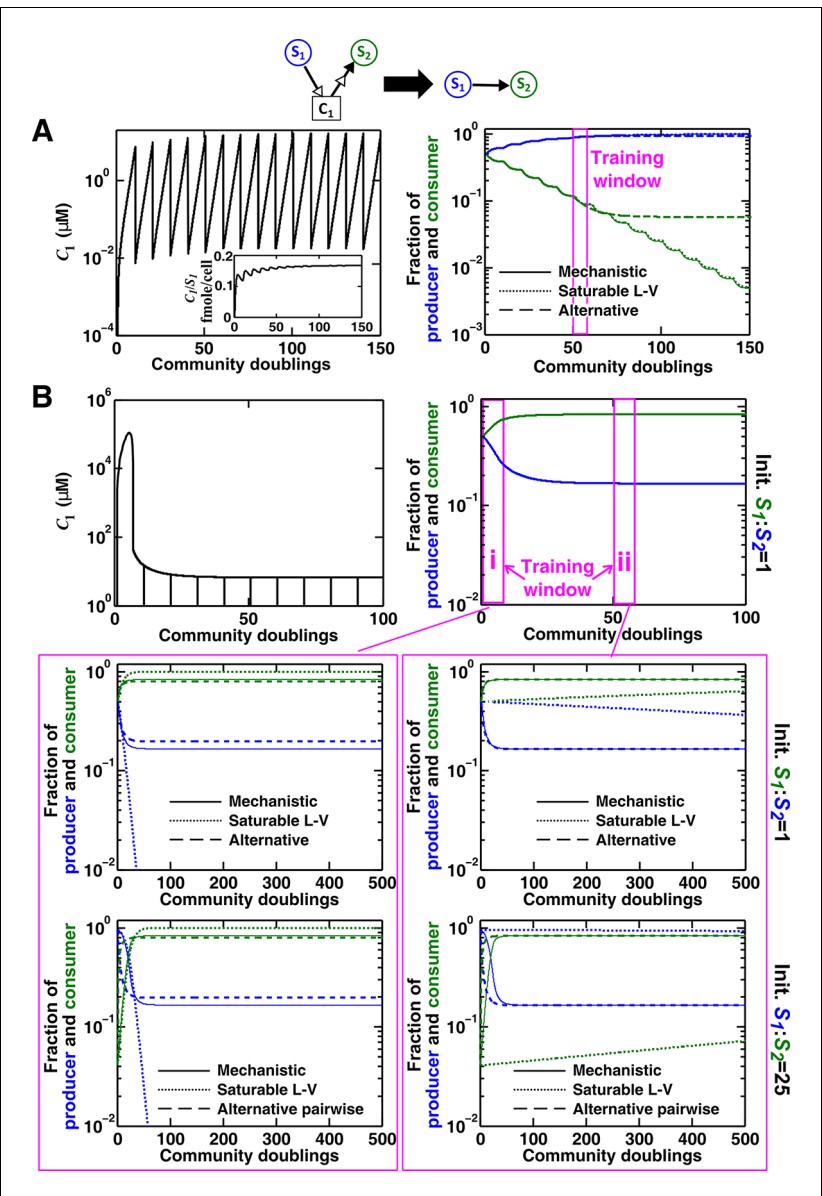

**Figure 4.** Saturable L-V and alternative pairwise models are not interchangeable. Consider a commensal community with a consumable mediator $C_1$. (A) The mediator accumulates without reaching a steady state within each dilution cycle as the consumer $S_2$ gradually goes extinct (*Figure 3B*, Case I). After a few tens of generations, $C_1$ becomes proportional to its producer density $S_1$ (inset in left panel). In this case, a saturable L-V (dotted) but not the alternative pairwise model (dashed) is suitable. All parameters are listed in *Figure 4—source data 1*. (B) The consumable mediator reaches a non-zero steady state within each dilution cycle (*Figure 3B*, Case II). From mechanistic dynamics where initial species ratio is 1, we use two training windows to derive saturable L-V (dotted) and alternative (dashed) pairwise models. We then use these pairwise models to predict dynamics of communities starting at two different ratios. The alternative model but not the saturable L-V predicts the mechanistic model dynamics. All parameters are listed in *Figure 4—source data 2*. Note that in all figures, population fractions (instead of population densities) are plotted, which fluctuate less during dilutions compared to mediator concentration.

The following source data is available for figure 4:

**Source data 1.** List of parameters for simulations in *Figure 4* on an interaction through a reusable mediator.
**Source data 2.** List of parameters for simulations in *Figure 4* on an interaction through a consumable mediator.

predictions from the alternative pairwise model are qualitatively wrong (*Figure 4A* right, compare dashed and solid lines). When a consumable mediator eventually reaches a non-zero steady state within each dilution cycle (*Figure 4B*, black), could a saturable L-V model still work? The saturable L-V model derived from training window **i** (initial 10 generations) fails to predict species coexistence regardless of initial species ratios (*Figure 4B* left magenta box, compare solid with dotted). In comparison, the saturable L-V model derived from training window **ii** (at steady-state species ratio) performs better, especially if the starting species ratio is identical to that of the training dynamics (*Figure 4B*, top panel in right magenta box). However, at a different starting species ratio, the saturable L-V model fails to predict which species dominates the community (*Figure 4B*, bottom panel in right magenta box). In contrast, community dynamics can be described by the alternative pairwise model derived from either window **i** or **ii** (*Figure 4B*, compare dashed and solid lines in left and right magenta boxes).

We have shown here that even when one species affects another species via a single mediator, either a saturable L-V model or an alternative pairwise model may be appropriate. The appropriate model depends on whether the mediator is reusable or consumable, how fitness of the two species compare, and initial species densities (*Figure 3*; *Figure 3—figure supplements 2–5*). Choosing the wrong pairwise model generates qualitatively flawed predictions (*Figure 4*). Considering that reusable and consumable mediators are both common in microbial interactions, our results call for revisiting the universality assumption of pairwise modeling.

## Two-mediator interactions require pairwise models different from single-mediator interactions

A species often affects another species via multiple mediators (*Kato et al., 2008*; *Yang et al., 2009*; *Traxler et al., 2013*; *Kim et al., 2013*). For example, a fraction of a population might die and release numerous chemicals, and some of these chemicals can simultaneously affect another individual. Here we examine the case where $S_1$ releases two reusable chemicals $C_1$ and $C_2$, both affecting the growth of $S_2$ (*Figure 5A*). Since the effect of each mediator can be described by a saturable L-V pairwise model (*Figure 3A*), we ask when the two mediators can be mathematically regarded as one mediator and described by a saturable L-V pairwise model (*Figure 5B*).

We assume that fitness effects from different chemical mediators on a focal species are additive. Not making this assumption will likely violate the additivity assumption essential to pairwise models. Additive fitness effects have been observed for certain 'homologous' metabolites. For example, in multi-substrate carbon-limited chemostats of *E. coli*, the fitness effects from glucose and galactose were additive (*Lendenmann and Egli, 1998*). 'Heterologous' metabolites such as carbon and nitrogen sources likely affect cell fitness in a multiplicative fashion. However, if $W_C$ and $W_N$, the fitness influences of released carbon and nitrogen with respect to those already in the environment, are both small (i.e. $W_C$, $W_N < < 1$), the additional relative fitness influence will be additive: $(1 + W_C)(1 + W_N) - 1 \approx W_C + W_N$. However, we need to keep in mind that even among homologous metabolites, fitness effects may not be additive (*Hermsen et al., 2015*). 'Sequential' metabolites (e.g. diauxic shift) provide another example of non-additivity. Similar to the previous section, we assume that all abiotic resources are unlimited.

For the two reusable mediators, depending on their relative 'potency' (defined in *Figure 5A* legend), their combined effect generally cannot be modeled as a single mediator except under special conditions (Methods-Conditions under which a saturable L-V pairwise model can represent one species influencing another via two reusable mediators). These special conditions include: (1) mediators share similar potency (*Figure 5—figure supplement 1B*), or (2) one mediator dominates the interaction (*Figure 5—figure supplement 1C*). If these conditions are not satisfied, we can easily find examples where saturable L-V pairwise models derived from a low-density community and from a high-density community have qualitatively different parameters (*Figure 5—figure supplement 1D*). Consequently, the future dynamics of a low-density community can be predicted by a saturable L-V model derived from a low-density community but not by a model derived from a high-density community (*Figure 5C*). Thus, even though each mediator can be modeled by saturable L-V, their joint effects may or may not be modeled by saturable L-V depending on the relative potencies of the two mediators and sometimes even on initial conditions (high or low initial $S_1$).

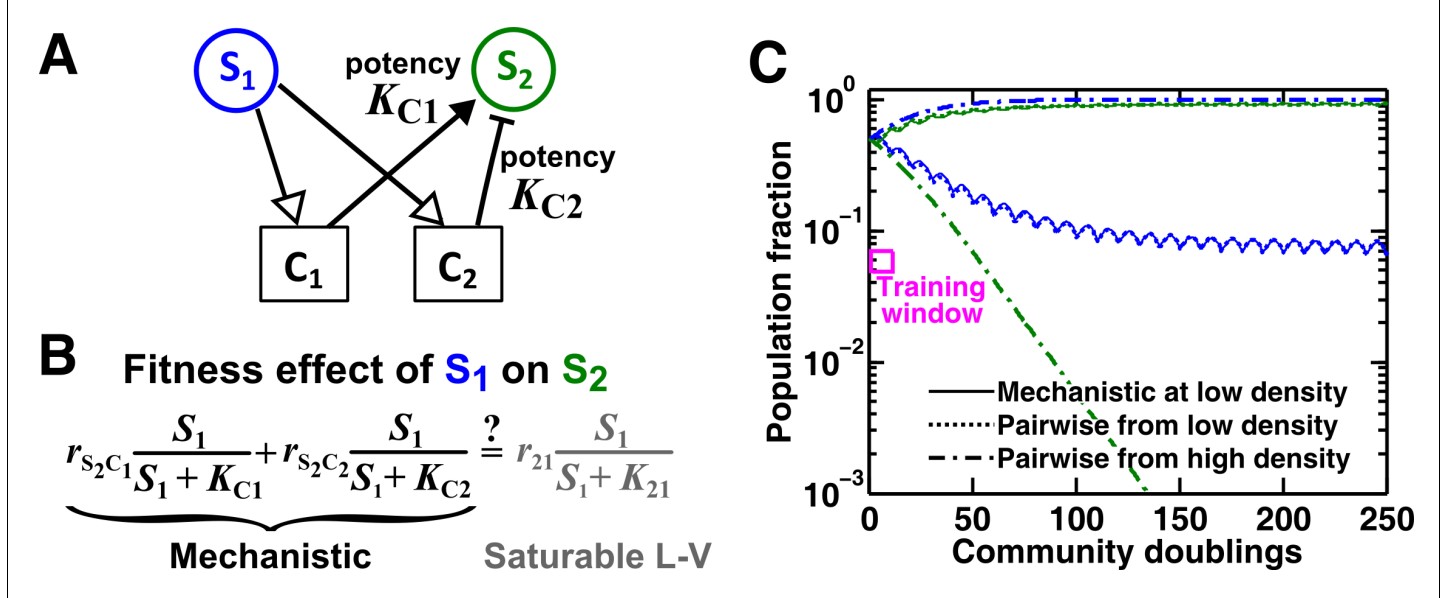

**Figure 5.** An example of a two-mediator interaction where a saturable L-V pairwise model may succeed or fail depending on initial conditions. (A) One species can affect another species via two reusable mediators, each with a different potency $K_{Ci}$ where $K_{Ci}$ is $K_{S_2 C_i} r_{10}/\beta_{C_i S_1}$ (Methods-Conditions under which a saturable L-V pairwise model can represent one species influencing another via two reusable mediators). A low $K_{Ci}$ indicates a strong potency (e.g. high release of $C_i$ by $S_1$ or low $C_i$ required to achieve half-maximal influence on $S_2$). (B) Under what conditions can an interaction via two reusable mediators with saturable effects on recipients be approximated by a saturable L-V pairwise model? (C) A community where the success or failure of a saturable L-V pairwise model depends on initial conditions. Here, $K_{C1} = 10^3$ cells/ml and $K_{C2} = 10^5$ cells/ml. Community dynamics starting at low $S_1$ (solid) can be predicted if the saturable L-V pairwise model is derived from reference dynamics starting at low (dotted). However, if we use a saturable L-V pairwise model derived from a community with high initial $S_1$, prediction is qualitatively wrong (dash dot line). See *Figure 5—figure supplement 1D* for an explanation why a saturable L-V pairwise model estimated at one community density may not be applicable to another community density. Simulation parameters are listed in *Figure 5—source data 1* .

The following source data and figure supplement are available for figure 5:

**Source data 1.** List of parameters for simulations in *Figure 5* on an interaction through two concurrent mediators.

**Source data 2.** List of parameters for simulations in *Figure 5—figure supplement 1* on an interaction through two concurrent mediators, assessed at high versus low cell densities.

**Figure supplement 1.** Except under special conditions, a pairwise interaction through two mediators may not be represented by a single saturable L-V model.

Similarly, when both mediators are consumable and do not accumulate (as in Cases II and III of *Figure 3B*), the fitness effect term becomes $\frac{r_{S_2 C_1} S_1}{\omega_{C_1} S_1 + \psi_{C_1} S_2} + \frac{r_{S_2 C_2} S_1}{\omega_{C_2} S_1 + \psi_{C_2} S_2}$. Except under special conditions (e.g. when $\omega_{C_1}$ and $\omega_{C_2}$ are zero, or when $\omega_{C_1}/\omega_{C_2} = \psi_{C_1}/\psi_{C_2}$, or when one mediator dominates the interaction), the two mediators may not be regarded as one. By the same token, when one mediator is a steady-state consumable and the other is reusable, they generally may not be regarded as a single mediator and would require yet a different pairwise model (i.e. with the fitness effect term $\frac{r_{S_2 C_1} S_1}{\omega_{C_1} S_1 + \psi_{C_1} S_2} + \frac{r_{S_2 C_2} S_1}{S_1 + K_{S_2 C_2} r_{10}/\beta_{C_2 S_1}}$).

In summary, when $S_1$ influences $S_2$ through multiple mediators, rarely can we approximate them as a single mediator. Sometimes, a pairwise model derived from one community may not apply to communities initiated at different densities (*Figure 5C*; *Figure 5—figure supplement 1D*). This casts further doubt on the usefulness of a single pairwise model for all pairwise microbial interactions.

## L-V competition model can fail if two competing species engage in an additional interaction

So far, by assuming that abiotic resources are always present in excess (e.g. in turbidostats), we have not considered species competition for abiotic resources. In this section, we consider a competitive commensal community in a batch environment where $S_1$ and $S_2$ compete for an essential shared resource $C_1$ supplied by the environment at a constant rate (e.g. constant light), and $S_1$ supplies an essential consumable metabolite $C_2$ to promote $S_2$ growth (*Figure 6A*, left). We show that an L-V pairwise model works for some but not all communities even though these communities qualitatively share the same interaction mechanism.

In our mechanistic model (Methods-Competitive commensal interaction, *Equation 47*), the fitness of $S_2$ is multiplicatively affected by $C_1$ and $C_2$ (*Mankad and Bungay, 1988*). We choose parameters such that the effect from $C_2$ to $S_2$ is far from saturation (e.g. linear with respect to $C_2$ and $S_1$) to simplify the problem. In our L-V pairwise model (*Figure 6A*, right; Methods-Competitive commensal interaction, *Equation 48*), intra- and inter-species competition is represented by the traditional logistic L-V model (*Equation 2*; *Gause, 1934*; *Thébault and Fontaine, 2010*; *Mougi and Kondoh, 2012*). We then introduce a linear term ($r_{21}S_1$) to describe the fitness effect of commensal interaction.

We tested various sets of mechanistic model parameters where the two species coexist in a steady fashion (*Figure 6B*), or one species goes extinct (*Figure 6C*), or species composition fluctuates (*Figure 6D*). L-V pairwise models deduced from a fixed period of training time could predict future dynamics in the first two cases, but failed to do so in the third case. Thus, depending on dynamic details of communities, a pairwise model sometimes works and sometimes fails.

To summarize our work, even for pairwise microbial interactions, depending on interaction mechanisms (reusable versus consumable mediator, single mediator versus multiple mediators), we will need to use a plethora of pairwise models to avoid qualitative failures in predicting which species dominates a community or whether species coexist (*Figures 3*, *4* and *5*). Sometimes, even when different communities share identical interaction mechanisms, depending on details such as relative species fitness, interaction strength, and initial conditions, the best-fitting pairwise model may or may not predict future dynamics (*Figure 3B*, *Figure 3—figure supplement 4*, *Figure 4*, *Figure 5*, *Figure 5—figure supplement 1*, and *Figure 6*). This defeats the very purpose of pairwise modeling – using a single equation form to capture fitness effects of all pairwise species interactions regardless of interaction mechanisms or quantitative details. In a community of more than two microbial species, interaction modification can cause pairwise models to fail (*Figure 7*). Even if species interact in an interaction chain and thus interaction modification does not occur, various chain segments may require different forms of pairwise models. Taken together, a pairwise model is unlikely to be effective for predicting community dynamics especially if interaction mechanisms are diverse.

## Discussions

Multispecies pairwise models are widely used in theoretical ecology due to their simplicity. These models assume that all pairwise species interactions can be captured by a single pairwise model regardless of interaction mechanisms or quantitative details of a community (universality assumption). This assumption may be satisfied if, for example, interaction mediators are always species themselves (e.g. prey-predation in a food web) so that pairwise models are equivalent to mechanistic models. However, interactions in microbial communities are diverse and often mediated by chemicals (*Figure 2*). Here, we consider the validity of universality assumption of pairwise models in well-mixed, two-species microbial communities. We have focused on various types of chemical-mediated interactions commonly encountered in microbial communities (*Figure 2*) (*Kato et al., 2005*; *Gause, 1934*; *Ghuysen, 1991*; *Jakubovics et al., 2008*; *Chen et al., 2004*; *D'Onofrio et al., 2010*; *Johnson et al., 1982*; *Hamilton and Ng, 1983*). For each type of species interaction, we construct a mechanistic model to generate reference community dynamics (akin to experimental results). We then attempt to derive the best-matching pairwise model and ask how predictive it is.

We first consider cases where abiotic resources are in excess. When one species affects another species via a single chemical mediator, either the saturable L-V or the alternative pairwise model is appropriate, depending on the interaction mechanism (consumable versus reusable mediator), relative fitness of the two species, and initial conditions (*Figure 3*; *Figure 3—figure supplement 2* to

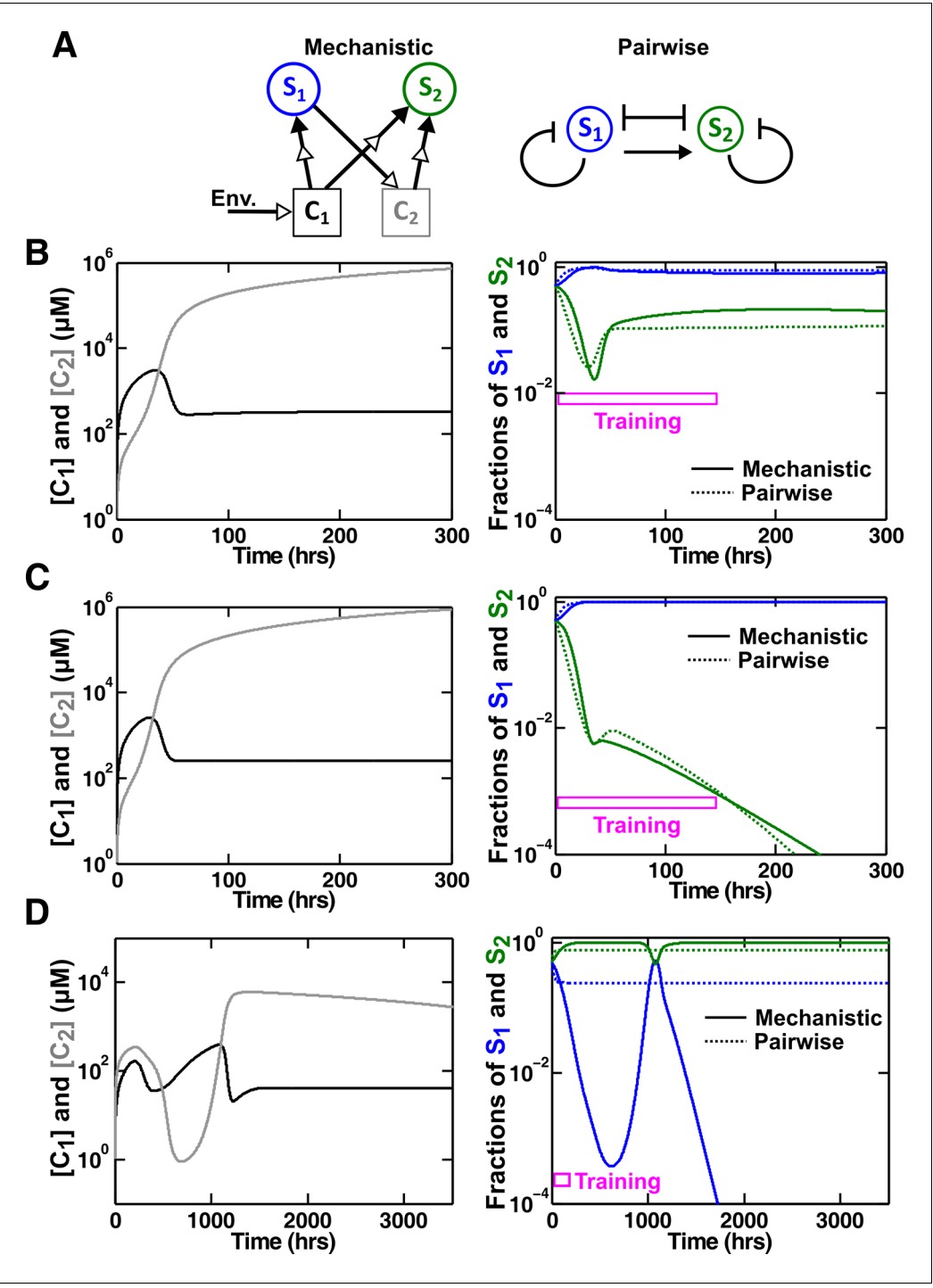

**Figure 6.** An example of a competitive commensal community where an L-V pairwise model may work or fail. (**A**) Left: Two species $S_1$ and $S_2$ compete for shared resource $C_1$. Additionally, $S_1$ produces $C_2$ that promotes the growth of $S_2$ upon consumption. Right: An L-V pairwise model captures the intra- and inter-species competition as well as the commensal interaction between the two species. (**B,C**) Examples where L-V pairwise models predict the mechanistic reference dynamics well. (**D**) An example where the L-V pairwise model fails to predict the dynamics qualitatively (note the much longer time range). Here, population fractions fluctuate due to changes in relative concentration of $C_1$ compared to $C_2$. In all cases, the pairwise model is derived from the population dynamics in the initial stages of growth (150 hr in all cases). Simulation parameters are listed in *Figure 6—source data 1*.

*Figure 6 continued on next page*

*Figure 6 continued*

The following source data is available for figure 6:

**Source data 1.** List of parameters for simulations in *Figure 6* on an interaction through a consumable mediator, for species consuming a shared abiotic resource.

*Figure 3—figure supplement 5*). These two models are not interchangeable (*Figure 4*). If one species influences another species through multiple mediators, then in general, these mediators may not be regarded as a single mediator nor representable by a single pairwise model. For example, for two reusable mediators, unless their potencies are similar or one mediator is much more potent than the other, saturable L-V model parameters can qualitatively differ depending on initial community density (*Figure 5—figure supplement 1D*). Consequently, a pairwise model derived from a high-density community generates false predictions for low-density communities (*Figure 5C*), limiting the usefulness of pairwise models. We then consider a community where two species compete for a shared resource while engaging in commensalism via a single chemical mediator. We find that the best-fitting L-V pairwise model can predict future dynamics in some but not all communities, depending on parameters used in the mechanistic model (*Figure 6*). Thus, although a single equation form can work in many cases, it generates qualitatively wrong predictions in many other cases.

In communities of more than two microbial species, indirect interactions via a third species can occur. When indirect interactions take the form of interaction chains, if each chain segment of two species engages in an independent interaction and can be represented by a pairwise model, then multispecies pairwise models can work (*Figure 7A-B*). However, as discussed above, pairwise equation forms may vary among chain segments depending on interaction mechanisms and quantitative details of a community. When indirect interactions take the form of interaction modification, even if each species pair can be accurately represented by a pairwise model, a multispecies pairwise model may fail (*Figure 7C–F*, ). Interaction modification includes trait modification (*Wootton, 2002*; *Werner and Peacor, 2003*; *Schmitz et al., 2004*), or, in our cases, mediator modification. Mediator modification is very common in microbial communities. For example, antibiotic released by one species to inhibit another species may be inactivated by a third species, and this type of indirect interactions can stabilize microbial communities (*Kelsic et al., 2015*; *Bairey et al., 2016*). As another example, interaction mediators are often generated by and shared among multiple species. For example in oral biofilms, organic acids such as lactic acid are generated from carbohydrate fermentation by many species (*Bradshaw et al., 1994*; *Marsh and Bradshaw, 1997*; *Kuramitsu et al., 2007*). Such by-products are also consumed by multiple species (*Kolenbrander, 2000*).

One can argue that an extended pairwise model (e.g. $\frac{dS_2}{dt} = r_{20}S_2 + \frac{r_{S_2}cS_1}{\varsigma + \omega S_1 + \psi S_2}S_2$) embodying both the saturable form and the alternative form can serve as a general-purpose model at least for pairwise interactions via a single mediator. In fact, even the effects of indirect interactions may be quantified and included in the model by incorporating higher-order interaction terms (*Case and Bender, 1981*; *Worthen and Moore, 1991*), although with many challenges (*Wootton, 2002*). In the end, although these strategies may lead to a sufficiently accurate phenomenological model especially within the training window, they may fail to predict future dynamics.

When might a pairwise model be useful? First, pairwise models have been instrumental in understanding ecological phenomena such as prey-predator oscillatory dynamics and coexistence of competing predator species (*Volterra, 1926*; *MacArthur, 1970*; *Case and Casten, 1979*; *Chesson, 1990*). In these cases, mechanistic models are either identical to pairwise models or can be transformed into pairwise models under simplifying assumptions. Second, pairwise models of pairwise species interactions can provide a bird's-eye view of strong or weak stimulatory or inhibitory interactions in a community. For example, *Vetsigian et al., 2011* found that interactions between soil-isolated *Streptomyces* strains are enriched for reciprocity – if A inhibits or promotes B, it is likely that B also inhibits or promotes A (*Vetsigian et al., 2011*). Third, pairwise models have been useful in qualitatively understanding species assembly rules in small communities (*Friedman et al., 2017*). That is, qualitative information regarding species survival in competitions among a small number of species may be used to predict survival in more diverse communities within a similar time window. Fourth, a pairwise model can serve as a starting point for generating hypotheses on species

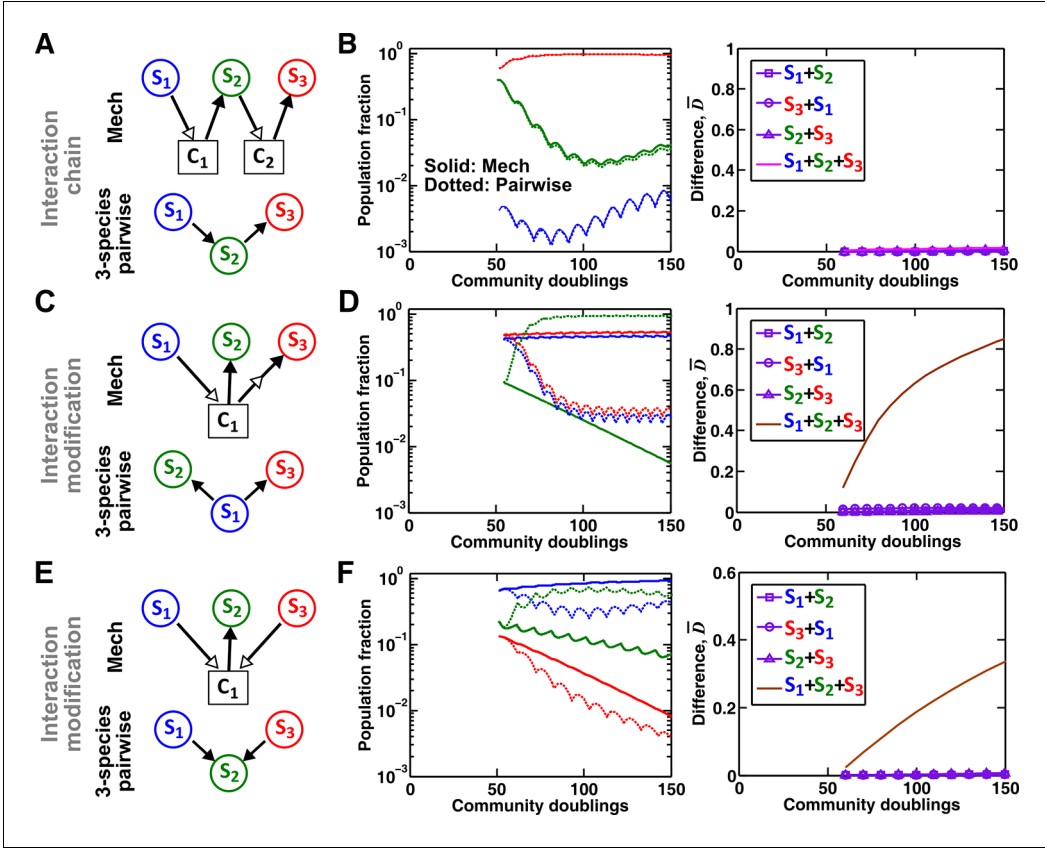

**Figure 7.** Interaction chain but not interaction modification may be represented by a multispecies pairwise model. We examine three-species communities engaging in indirect interactions. Each species pair is representable by a two-species pairwise model (saturable L-V or alternative pairwise model, purple in the right columns of **B**, **D**, and **F**). We then use these two-species pairwise models to construct a three-species pairwise model, and test how well it predicts the dynamics known from the mechanistic model. In **B**, **D**, and **F**, left panels show dynamics from the mechanistic models (solid lines) and three-species pairwise models (dotted lines). Right panels show the difference metric $\bar{D}$. (**A–B**) Interaction chain: $S_1$ affects $S_2$, and $S_2$ affects $S_3$. The two interactions employ independent mediators $C_1$ and $C_2$, and both interactions can be represented by the saturable L-V pairwise model. The three-species pairwise model matches the mechanistic model in this case. Simulation parameters are provided in *Figure 7—source data 1*. (**C–F**) Interaction modification. In both cases, the three-species pairwise model fails to predict reference dynamics even though the dynamics of each species pair can be represented by a pairwise model. (**C–D**) $S_3$ consumes $C_1$, a mediator by which $S_1$ stimulates $S_2$. Parameters are listed in *Figure 7—source data 2*. Here, $S_1$ changes the nature of interaction between $S_2$ and $S_3$: $S_2$ and $S_3$ do not interact in the absence of $S_1$, but $S_3$ inhibits $S_2$ in the presence of $S_1$. The three-species pairwise model makes qualitatively wrong prediction about species coexistence. As expected, if $S_3$ does not remove $C_1$, the three-species pairwise model works (*Figure 7—figure supplement 1A–B*). (**E–F**) $S_1$ and $S_3$ both supply $C_1$ which stimulates $S_2$. Here, no species changes 'the nature of interactions' between any other two species: both $S_1$ and $S_3$ contribute reusable $C_1$ to stimulate $S_2$. $S_1$ promotes $S_2$ regardless of $S_3$; $S_3$ promotes $S_2$ regardless of $S_1$; $S_1$ and $S_3$ do not interact regardless of $S_2$. However, a multispecies pairwise model assumes that the fitness effects from the two producers on $S_2$ will be additive, whereas in reality, the fitness effect on $S_2$ saturates at high . As a result, the three-species pairwise model qualitatively fails to capture relative species abundance. As expected, if $C_1$ affects $S_2$ in a linear fashion, the community dynamics is accurately captured in the multispecies pairwise model (*Figure 7—figure supplement 1C–D*). Simulation parameters are listed in *Figure 7—source data 3*.

The following source data and figure supplement are available for figure 7:

**Source data 1.** List of parameters for simulations in *Figure 7B* on interaction between three species in a chain.

**Source data 2.** List of parameters for simulations in *Figure 7D* on interaction modification through consumption of a shared mediator by a third species.

*Figure 7 continued on next page*

*Figure 7 continued*

**Source data 3.** List of parameters for simulations in *Figure 7F* on interaction modification through production of a shared mediator by a third species.

**Source data 4.** List of parameters for simulations in *Figure 7—figure supplement 1B* on an interaction between three species through a shared reusable mediator affecting multiple species.

**Source data 5.** List of parameters for simulations in *Figure 7—figure supplement 1D* on an interaction between three species through a shared reusable mediator produced by multiple species.

**Figure supplement 1.** A multispecies pairwise model can work under special conditions.

interactions (e.g. *Li et al., 2015*). Note that when applied to microbial communities (*Mounier et al., 2008*; *Stein et al., 2013*; *Marino et al., 2014*), a fitting pairwise model means that the training dynamics of the community under investigation can be approximated by a theoretical community where species interactions satisfy the additivity and universality assumptions of pairwise models. Even though the theoretical community is likely different from the real community, hypothesis formulation can still be valuable. Finally, pairwise models can be useful in making predictions of limited scales. For example, Stein et al. used 2/3 of community dynamics data as a training set to derive a multispecies pairwise model, and in the best-case scenario, the model generated reasonable predictions on the remaining 1/3 of data (*Stein et al., 2013*). However, as we have shown, pairwise models can generate qualitatively wrong predictions (*Figures 4–7*), especially if interaction mechanisms are diverse such as in microbial communities. Not surprisingly, predicting qualitative consequences of species removal or addition using a pairwise model has encountered difficulties, especially in communities of more than three species (*Mounier et al., 2008*; *Friedman et al., 2017*).

An alternative to a pairwise model is a mechanistic model. How much information about interaction mechanisms do we need to construct a mechanistic model? That is, what is the proper level of abstraction which captures the phenomena of interest, yet avoids unnecessary details (*Li et al., 2015*; *Durrett and Levin, 1994*)? For example, Tilman argued that if a small number of mechanisms (e.g. the 'axes of trade-offs' in species traits) could explain much of the observed pattern (e.g. species coexistence), then this abstraction would be highly revealing (*Tilman, 1987*). However, the choice of abstraction is often not obvious. Consider for example a commensal community where $S_1$ grows exponentially (not explicitly depicted in equations in *Figure 8*) and the net growth rate of $S_2$, which is normally zero, is promoted by mediator $C$ from $S_1$ in a linear fashion (*Figure 8*). If we do not know how $S_1$ stimulates $S_2$, we can still construct an L-V pairwise model (*Figure 8A*). If we know the identity of mediator $C$ and realize that $C$ is consumable, then we can instead construct a mechanistic model incorporating $C$ (*Figure 8B*). However, if $C$ is produced from a precursor via an enzyme $E$ released by $S_1$, then we get a different form of mechanistic model (*Figure 8C*). If, on the other hand, $E$ is anchored on the membrane of $S_1$ and each cell expresses a similar amount of $E$, then equations in *Figure 8D* are mathematically equivalent to *Figure 8B*. This simple example, inspired by extracellular breakdown of cellulose into a consumable sugar $C$ (*Bayer and Lamed, 1986*; *Felix and Ljungdahl, 1993*; *Schwarz, 2001*), illustrates how knowledge of mechanisms may eventually help us determine the right level of abstraction.

In summary, under certain circumstances, we may already know that microbial interaction mechanisms fall within the domain of validity for a particular pairwise model. In these cases, a pairwise model provides the appropriate level of abstraction, and constructing such a pairwise model is much easier than a mechanistic model (*Figure 1*). However, if we do not know whether a pairwise model is valid, we will need to be cautious since pairwise models can fail to even qualitatively capture pairwise microbial interactions. We need to be equally careful when extrapolating and generalizing conclusions obtained from pairwise models, especially for communities where species interaction mechanisms are diverse. Considering recent advances in identifying and quantifying interactions, we advocate a transition to models that incorporate interaction mechanisms at the appropriate level of abstraction.

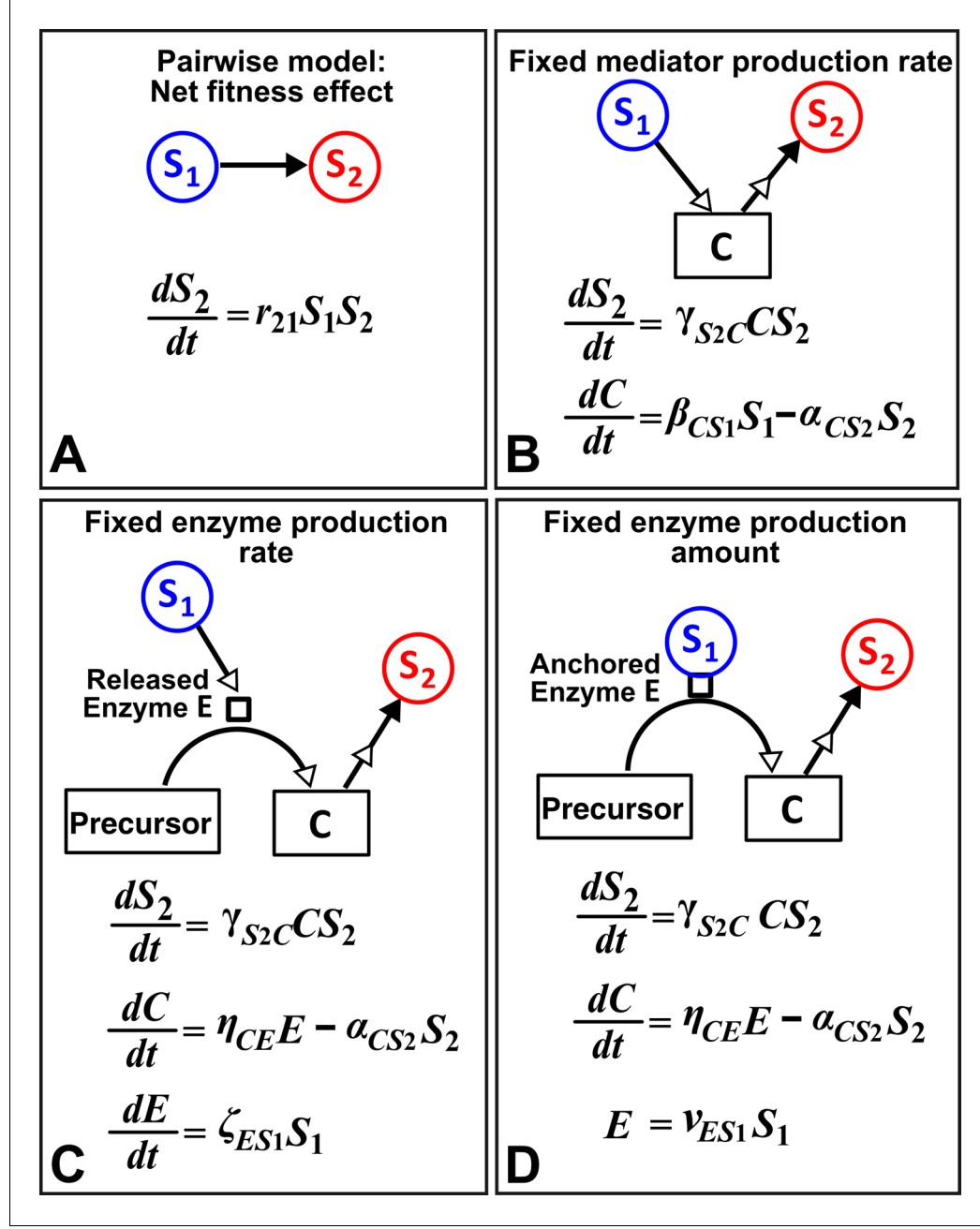

**Figure 8.** Different levels of abstraction in a mechanistic model. How one species ($S_1$) may influence another ($S_2$) can be mechanistically modeled at different levels of abstraction. For simplicity, here we assume that interaction strength scales in a linear (instead of saturable) fashion with respect to mediator concentration or species density. The basal fitness of $S_2$ is zero. (A) In the simplest form, $S_1$ stimulates $S_2$ in an L-V pairwise model. (B) In a mechanistic model, we may realize that $S_1$ stimulates $S_2$ via a mediator C which is consumed by $S_2$. The corresponding mechanistic model is given. (C) Upon probing more deeply, it may become clear that $S_1$ stimulates $S_2$ via an enzyme E, where E degrades an abundant precursor (such as cellulose) to generate mediator C (such as glucose). In the corresponding mechanistic model, we may assume that E is released by $S_1$ at a rate $\zeta_{ES1}$ and that E liberates C at a rate $\eta_{CE}$. (D) If instead E is anchored on the cell surface (e.g. cellulosome), then *E* is proportional to $S_1$. If we substitute E into the second equation, then (B) and (D) become equivalent. Thus, when enzyme is anchored on cell surface but not when enzyme is released, the mechanistic knowledge of enzyme can be neglected.

## Materials and methods

### Interaction modification but not interaction chain violates the additivity assumption

In a pairwise model, the fitness of a focal species $S_i$ is the sum of its 'basal fitness' ($r_{i0}$, the net growth rate of a single $\underline{S_i}$ individual in the absence of any intra-species or inter-species interactions) and the additive fitness effects exerted by pairwise interactions with other members of the community. Mathematically, an $N$-species pairwise model is often formulated as

$$\frac{dS_i}{dt} = \left( r_{i0} + \sum_{j=1}^{N} f_{ij}(S_j) \right) S_i \tag{6}$$

Here, $f_{ij}(S_j)$ describes how $S_j$, the density of species $S_j$, positively or negatively affects the fitness of $S_i$, and is a linear or nonlinear function of only $S_j$.

Indirect interactions via a third species fall under two categories (*Wootton, 1993*). The first type is known as 'interaction chain' or 'density-mediated indirect interactions'. For example, the consumption of plant $S_1$ by herbivore $S_2$ is reduced when the density of herbivore is reduced by carnivore $S_3$. In this case, the three-species pairwise model

$$\begin{cases} \frac{dS_1}{dt} &= (r_{10} - f_{12}(S_2))S_1 \\ \frac{dS_2}{dt} &= (r_{20} + f_{21}(S_1) - f_{23}(S_3))S_2 \\ \frac{dS_3}{dt} &= (r_{30} + f_{32}(S_2))S_3 \end{cases} \tag{7}$$

does not violate the additivity assumption (compare with *Equation 6* (*Case and Bender, 1981*; *Wootton, 1994*).

The second type of indirect interactions is known as 'interaction modification' or 'trait-mediated indirect interactions' or 'higher order interactions' (*Vandermeer, 1969*; *Wootton, 1994*; *Billick and Case, 1994*; *Wootton, 2002*), where a third species modifies the 'nature of interaction' from one species to another (*Wootton, 2002*; *Werner and Peacor, 2003*; *Schmitz et al., 2004*). For example, when carnivore is present, herbivore will spend less time foraging and consequently plant density increases. In this case, $f_{12}$ in *Equation 7* is a function of both $S_2$ and $S_3$, violating the additivity assumption.

### Summary of simulation files

Simulations are based on Matlab and executed on an ordinary PC. Steps are:

Step 1: Identify monoculture parameters $r_{i0}$, $r_{ii}$, and $K_{ii}$ (*Figure 1—figure supplement 2C*, Row 1 and Row 2).

Step 2: Identify interaction parameters $r_{ij}$, $r_{ji}$, $K_{ij}$, and $K_{ji}$ where $i \neq j$ (*Figure 1—figure supplement 2C*, Row 3).

Step 3: Calculate distance $\bar{D}$ between population dynamics of the reference mechanistic model and the approximate pairwise model over a period of time outside of the training window to assess if the pairwise model is predictive.

Fitting is performed using nonlinear least squares (lsqnonlin routine) with default optimization parameters. The following list describes the m-files used for different steps of the analysis:

| File name | Function |
| --- | --- |
| FitCost_BasalFitness<br>*Source code 1* | Calculates the cost function for monocultures (i.e. the difference between the target mechanistic model dynamics and the dynamics obtained from the pairwise model) |
| FitCost_BFSatLV.m<br>*Source code 2* | Calculates the cost function for communities (i.e. the difference between the target mechanistic model dynamics and the dynamics obtained from the saturable L-V pairwise model) |
| FitCost_BFSatLV_Dp.m<br>*Source code 3* | Calculates the cost function for communities (i.e. the difference between the target mechanistic model dynamics and the dynamics obtained from the alternative pairwise model) |
| DynamicsMM_WM_MonocultureDpMM.m<br>*Source code 4* | Returns growth dynamics for monocultures, based on the mechanistic model |

| | |
|---|---|
| DynamicsMMSS_WM_NetworkDpMM.m **Source code 5** | Returns growth dynamics for communities of multiple species, based on the mechanistic model |
| DynamicsWM_NetworkBFSatLV.m **Source code 6** | Returns growth dynamics for communities of multiple species, based on the saturable L-V pairwise model |
| DynamicsWM_NetworkBFSatLV_Dp.m **Source code 7** | Returns growth dynamics for communities of multiple species, based on the alternative pairwise model |
| DeriveBasalFitnessMM_WM_DpMM.m **Source code 8** | Estimates monoculture parameters of pairwise model (Step 1) |
| DeriveBFSatLVMMSS_WM_DpMM.m **Source code 9** | Estimates saturable L-V pairwise model interaction parameters (Step 2) |
| DeriveBFSatLVMMSS_WM_DpMM_Dp.m **Source code 10** | Estimates alternative pairwise model interaction parameters (Step 2) |
| DeriveBFSatLVMMSS_WM_DpMM_r21.m **Source code 11** | Estimates saturable L-V pairwise model interaction parameters ($r_{21}$ and $K_{21}$) in cases where we know that $S_2$ is only affected by $S_1$, to accelerate optimization |
| DeriveBFSatLVMM_WM_DpMM_Dp_r21.m **Source code 12** | Estimates alternative pairwise model interaction parameter ($r_{21}$) in cases where we know that $S_2$ is only affected by $S_1$ and that $K_{S2C1} = K_{C1S2}$ to accelerate optimization |
| DynamicsWM_NetworkBFLogLV_DI.m **Source code 13** | Returns growth dynamics for communities of two species competing for an environmental resource while engaging in an additional interaction, based on the logistic L-V pairwise model (**Figure 6**) |
| C2Sp2_ARCLi_NoSatDp_FitBFLogLV_DI.m **Source code 14** | Estimates logistic L-V pairwise model interaction parameters for communities of two species competing for an environmental resource while engaging in an additional interaction, and compares community dynamics from pairwise and mechanistic models (**Figure 6**) |
| Dynamics_WM_NetworkDpMM_ODE23.m **Source code 15** | Defines differential equations when using Matlab's ODE23 solver to calculate community dynamics |
| Case_C1Sp2_CmnsDp_ODE23.m **Source code 16** | Example of using Matlab ODE23 solver for calculating community dynamics |

## Deriving a pairwise model for interactions mediated by a single consumable mediator

To facilitate mathematical analysis, we assume that requirements calculated below are eventually satisfied within each dilution cycle (see **Figure 3—figure supplement 4** for an example where dilution cycles necessitated by long convergence time violate requirements for a pairwise model to converge to the mechanistic model). We further assume that $r_{10} > 0$ and $r_{20} > 0$ so that species cannot go extinct in the absence of dilution. See **Figure 3—figure supplement 5** for a summary of this section.

When $S_1$ releases a consumable mediator which stimulates the growth of $S_2$, the mechanistic model as per **Figure 3B**, is

$$\begin{cases} \frac{dS_1}{dt} = r_{10} S_1 \\ \frac{dS_2}{dt} = r_{20} S_2 + r_{S_2 C_1} \frac{C_1}{C_1 + K_{S_2 C_1}} S_2 \\ \frac{dC_1}{dt} = \beta_{C_1 S_1} S_1 - \alpha_{C_1 S_2} \frac{C_1}{C_1 + K_{C_1 S_2}} S_2 = \left( \beta_{C_1 S_1} - \alpha_{C_1 S_2} \frac{C_1}{C_1 + K_{C_1 S_2}} \frac{S_2}{S_1} \right) S_1 \end{cases} \quad (8)$$

Let $C_1(t=0) = C_{10} = 0$; $S_1(t=0) = S_{10}$; and $S_2(t=0) = S_{20}$. Note that the initial condition $C_{10} = 0$ can be easily imposed experimentally by pre-washing cells. Under which conditions can we eliminate $C_1$ so that we can obtain a pairwise model of $S_1$ and $S_2$?

Define $R_S = S_2/S_1$ as the ratio of the two populations.

$$\begin{aligned} \frac{dR_S}{dt} &= \frac{\frac{dS_2}{dt} S_1 - S_2 \frac{dS_1}{dt}}{S_1^2} = \left( r_{20} + r_{S_2 C_1} \frac{C_1}{C_1 + K_{S_2 C_1}} \right) \frac{S_2}{S_1} - \frac{S_2}{S_1^2} r_{10} S_1 \\ &= \left( r_{20} + r_{S_2 C_1} \frac{C_1}{C_1 + K_{S_2 C_1}} - r_{10} \right) R_S \end{aligned} \quad (9)$$

## Case I: $r_{10} - r_{20} > r_{S_2 C_1}$

Since producer $S_1$ always grows faster than consumer $S_2$, $R_S \to 0$ as $t \to \infty$. Define $\tilde{C}_1 = C_1/S_1$ ('~' indicating scaling against a function).

$$\frac{d\tilde{C}_1}{dt} = \frac{d(C_1/S_1)}{dt} = \frac{\frac{dC_1}{dt}S_1 - C_1\frac{dS_1}{dt}}{S_1^2} = \frac{\left(\beta_{C_1 S_1}S_1 - \frac{\alpha_{C_1 S_2}C_1}{C_1 + K_{C_1 S_2}}S_2\right)S_1 - C_1 r_{10}S_1}{S_1^2} \tag{10}$$

$$= \beta_{C_1 S_1} - r_{10}\tilde{C}_1 - \frac{\alpha_{C_1 S_2}\tilde{C}_1}{\tilde{C}_1 + K_{C_1 S_2}\exp(-r_{10}t)/S_{10}}R_S$$

Since $R_S$ declines exponentially with a rate faster than $|r_{20} + r_{S_2 C_1} - r_{10}|$, we can ignore the third term of the right hand side of *Equation 10* if it is much smaller than the first term. That is,

$$\frac{\alpha_{C_1 S_2}\tilde{C}_1}{\tilde{C}_1 + K_{C_1 S_2}\exp(-r_{10}t)/S_{10}}R_S < \alpha_{C_1 S_2}R_S \le \alpha_{C_1 S_2}R_S(0)\exp(-|r_{20} + r_{S_2 C_1} - r_{10}|t) \ll \beta_{C_1 S_1}.$$

Thus for $t \gg \ln\left(\frac{\alpha_{C_1 S_2}R_S(0)}{\beta_{C_1 S_1}}\right) \Big/ |r_{20} + r_{S_2 C_1} - r_{10}|$, $\frac{d\tilde{C}_1}{dt} \approx \beta_{C_1 S_1} - r_{10}\tilde{C}_1$. When initial $\tilde{C}_1$ is 0, this equation can be solved to yield: $\tilde{C}_1 \approx \beta_{C_1 S_1}(1 - \exp(-r_{10}t))/r_{10}$. After time of the order of $1/r_{10}$, the second term can be neglected. Thus, $\tilde{C}_1 \approx \beta_{C_1 S_1}/r_{10}$ after time of the order of $\max\left(\ln\left(\frac{\alpha_{C_1 S_2}R_S(0)}{\beta_{C_1 S_1}}\right)\Big/|r_{20} + r_{S_2 C_1} - r_{10}|, 1/r_{10}\right)$. Then $C_1$ can be replaced by $(\beta_{C_1 S_1}/r_{10})S_1$ in *Equation 8*, and a saturable L-V pairwise model can be derived.

## Case II: $r_{S_2 C_1} > r_{10} - r_{20} > 0$

For *Equation 8*, we find that a steady state solution for $C_1$ and $R_S$, denoted respectively as $C_1^*$ and $R_S^*$, exist. They can be easily found by setting the growth rates of **S₁** and **S₂** to be equal, and $dC_1/dt$ to zero.

$$\begin{cases} C_1^* = \frac{r_{10} - r_{20}}{r_{20} + r_{S_2 C_1} - r_{10}}K_{S_2 C_1} \\ R_S^* = \frac{\beta_{C_1 S_1}}{\alpha_{C_1 S_2}}\left(1 + \frac{K_{C_1 S_2}}{C_1^*}\right) \end{cases} \tag{11}$$

However, if $C_1$ has not yet reached steady state, imposing steady state assumption would falsely predict $R_S$ at steady state and thus remaining at its initial value (*Figure 4Bii*, dotted lines). Since $dC_1/dt$ in *Equation 8* is the difference between two exponentially growing terms, we factor out the exponential term $S_1$ to obtain

$$\frac{dC_1}{dt} = \left(\beta_{C_1 S_1} - \alpha_{C_1 S_2}\frac{C_1}{C_1 + K_{C_1 S_2}}\frac{S_2}{S_1}\right)S_1 = \beta_{C_1 S_1}f(C_1, R_S)S_1 \tag{12}$$

where $f(C_1, R_S) = 1 - \frac{\alpha_{C_1 S_2}}{\beta_{C_1 S_1}}\frac{C_1}{C_1 + K_{C_1 S_2}}R_S$. When $f \approx 0$, we can eliminate $C_1$ and obtain an alternative pairwise model

$$\frac{dS_2}{dt} = r_{20}S_2 + r_{S_2 C_1}\frac{\beta_{C_1 S_1}K_{C_1 S_2}S_1}{\beta_{C_1 S_1}(K_{C_1 S_2} - K_{S_2 C_1})S_1 + \alpha_{C_1 S_2}K_{S_2 C_1}S_2}S_2 \tag{13}$$

Or

$$\frac{dS_2}{dt} = r_{20}S_2 + \frac{r_{S_2 C_1}S_1}{\omega S_1 + \psi S_2}S_2 \tag{4}$$

where $\omega$ and $\psi$ are constants (*Figure 3Bii*).

For certain conditions (which will be discussed at the end of this section, *Figure 3—figure supplement 5A*), this alternative model can make reasonable predictions of community dynamics even before the community reaches the steady state (*Figure 4Bii*, compare dashed and solid lines). Below we discuss the general properties of community dynamics and show that there exists a time scale $t_f$ after which it is reasonable to assume $f \approx 0$ and the alternative model can be derived. We also estimate $t_f$ for several scenarios.

We first make $C_1$ and $R_S$ dimensionless by defining $\hat{C}_1 = C_1/C_1^*$ and $\hat{R}_S = R_S/R_S^*$ ('^' indicating scaling against steady state values). *Equation 9* can then be rewritten as

$$\frac{d\hat{R}_S}{dt} = \left(r_{20} + r_{S_2C_1}\frac{\hat{C}_1}{\hat{C}_1 + \hat{K}_{S_2C_1}} - r_{10}\right)\hat{R}_S \tag{14}$$

where $\hat{K}_{S_2C_1} = K_{S_2C_1}/C_1^*$.

From *Equations 8 and 11*, we obtain

$$\frac{d\frac{C_1}{C_1^*}}{dt} = \frac{1}{C_1^*}\left(\beta_{C_1S_1} - \frac{\alpha_{C_1S_2C_1}}{C_1 + K_{C_1S_2}}\frac{R_S}{R_S^*}R_S^*\right)S_1 = \frac{1}{C_1^*}\left(\beta_{C_1S_1} - \frac{\alpha_{C_1S_2C_1}}{C_1 + K_{C_1S_2}}\hat{R}_S R_S^*\right)S_1$$

$$= \frac{1}{C_1^*}\left(\beta_{C_1S_1} - \frac{\alpha_{C_1S_2C_1}/C_1^*}{C_1/C_1^* + K_{C_1S_2}/C_1^*}\hat{R}_S\frac{\beta_{C_1S_1}}{\alpha_{C_1S_2}}\left(1 + \frac{K_{C_1S_2}}{C_1^*}\right)\right)S_1$$

$$= \frac{1}{C_1^*}\beta_{C_1S_1}\left(1 - \frac{\hat{C}_1\left(1 + \hat{K}_{C_1S_2}\right)}{\hat{C}_1 + \hat{K}_{C_1S_2}}\hat{R}_S\right)S_1$$

or

$$\frac{d\hat{C}_1}{dt} = \hat{\beta}_{C_1S_1}\left[1 - \frac{\hat{C}_1\left(1 + \hat{K}_{C_1S_2}\right)}{\hat{C}_1 + \hat{K}_{C_1S_2}}\hat{R}_S\right]S_1 \tag{15}$$

where $\hat{\beta}_{C_1S_1} = \beta_{C_1S_1}/C_1^*$ and $\hat{K}_{C_1S_2} = K_{C_1S_2}/C_1^*$.

Using these scaled variables, $f$ (i.e. the square bracket in *Equation 15*) can be rewritten as

$$f(\hat{C}_1, \hat{R}_S) = 1 - \frac{\hat{C}_1\left(1 + \hat{K}_{C_1S_2}\right)}{\hat{C}_1 + \hat{K}_{C_1S_2}}\hat{R}_S \tag{16}$$

and

$$\frac{d\hat{C}_1}{dt} = \hat{\beta}_{C_1S_1}f(\hat{C}_1, \hat{R}_S)S_1 \tag{17}$$

*Equations 14 and 17* allow us to construct a phase portrait where the x axis is $\hat{C}_1$ and the y axis is $\hat{R}_S$ (*Figure 3—figure supplement 2A–D*). Note that at steady state, $(\hat{C}_1, \hat{R}_S) = (1,1)$. Setting *Equation 16* to zero:

$$\hat{R}_S = \left(1 + \hat{K}_{C_1S_2}/\hat{C}_1\right)/\left(1 + \hat{K}_{C_1S_2}\right) \text{ or } \hat{C}_1 = \hat{K}_{C_1S_2}/\left[\hat{R}_S\left(1 + \hat{K}_{C_1S_2}\right) - 1\right] \tag{18}$$

defines the $f$-zero-isocline on the $\hat{C}_1 - \hat{R}_s$ phase plane (i.e. values of $(\hat{C}_1, \hat{R}_S)$ at which $f(\hat{C}_1, \hat{R}_S) = 0$ and thus $\hat{C}_1$ can be eliminated to obtain a pairwise model; *Figure 3—figure supplement 2A–D* blue lines). As shown in *Figure 3—figure supplement 2A*, the phase portrait is divided into four regions by the $f$-zero-isocline (blue) and the steady state $\hat{C}_1 = 1$ (vertical solid line), and grey arrows dictate the direction of the community dynamics trajectory $(\hat{C}_1, \hat{R}_S)$. Starting from 'initial state' ($\hat{C}_1(t = 0) = 0$, $\hat{R}_S(t = 0)$), the trajectory moves downward right (brown circles and orange lines in *Figure 3—figure supplement 2A–D*) until it hits $\hat{C}_1 = 1$. Then, it moves upward right and eventually hits the $f$-zero-isocline. Afterward, the trajectory moves toward the steady state (green circles) very closely along (and not superimposing) the $f$-zero-isocline during which the alternative pairwise model can be derived (*Figure 3—figure supplement 2A–D*).

It is difficult to solve *Equations 14 and 15* analytically because the detailed community dynamics depends on the parameters and the initial species composition in a complicated way. However, under certain initial conditions, we can estimate $t_f$, the time scale for the community to approach the $f$-zero-isocline. Note that $t_f$ is not a precise value. Instead it estimates the acclimation time scale after which a pairwise model can be derived.

One assumption used when estimating all $t_f$ is that $S_{10}$ is sufficiently high (*Figure 3—figure supplement 5B*) to avoid the long lag phase that is otherwise required for the mediator to accumulate to a high enough concentration.

From *Equation 18*, the asymptotic value for the $f$-zero-isocline is

$$\hat{R}_S(\hat{C}_1 \to \infty) = 1/(1 + \hat{K}_{C_1 S_2}) \tag{19}$$

This is plotted as a black dotted line in *Figure 3—figure supplement 2A–D*.

Below we consider three different initial conditions for $\hat{R}_S(t = 0)$:

## Case II-1. $\hat{R}_S(t=0) \gg \max(1, \hat{K}_{S_2 C_1}^{-1})$

From *Equation 11*, this becomes $R_S(0)/R_S^* \gg \max(1, \frac{r_{10} - r_{20}}{r_{20} + r_{S_2 C_1} - r_{10}})$.

A typical trajectory of the system is shown in *Figure 3—figure supplement 2B*: at time $t = 0$, using *Equations 14 and 15*, the community dynamics trajectory (orange solid line in *Figure 3—figure supplement 2B* inset) has a slope of

$$\left.\frac{d\hat{R}_S}{d\hat{C}_1}\right|_{\hat{C}_1(0)=0} = \left.\frac{d\hat{R}_S/dt}{d\hat{C}_1/dt}\right|_{t=0} = \frac{(r_{20} - r_{10})\hat{R}_S(0)}{\hat{\beta}_{C_1 S_1} S_1(0)} \tag{20}$$

From *Equation 18*, the slope of the $f$-zero-isocline (blue line in *Figure 3—figure supplement 2B* inset) at $\hat{R}_S = \hat{R}_S(0)$ is

$$
\begin{aligned}
\left.\frac{d\hat{R}_S}{d\hat{C}_1}\right|_{\hat{R}_S=\hat{R}_S(0)} &= \left.\frac{d\left[(1+\hat{K}_{C_1 S_2}/\hat{C}_1)/(1+\hat{K}_{C_1 S_2})\right]}{d\hat{C}_1}\right|_{\hat{R}_S=\hat{R}_S(0)} \\
&= \left.\frac{-1}{1+\hat{K}_{C_1 S_2}}\frac{\hat{K}_{C_1 S_2}}{\hat{C}_1^2}\right|_{\hat{R}_S=\hat{R}_S(0)} = \frac{-\hat{K}_{C_1 S_2}}{1+\hat{K}_{C_1 S_2}}\left(\frac{\hat{R}_S(0)(1+\hat{K}_{C_1 S_2})-1}{\hat{K}_{C_1 S_2}}\right)^2 \\
&= \frac{-\left[(1+\hat{K}_{C_1 S_2})\hat{R}_S(0)-1\right]^2}{(1+\hat{K}_{C_1 S_2})\hat{K}_{C_1 S_2}} \approx \frac{-(1+\hat{K}_{C_1 S_2})\hat{R}_S(0)^2}{\hat{K}_{C_1 S_2}}
\end{aligned}
\tag{21}
$$

The approximation in the last step is due to the very definition of Case II-1: $\hat{R}_S(t=0) \gg 1$. The initial steepness of the community dynamics trajectory (*Equation 20*) will be much smaller than that of the $f$-zero-isocline (*Equation 21*) if

$$S_1(0) \gg \frac{\hat{K}_{C_1 S_2}(r_{10}-r_{20})}{\hat{\beta}_{C_1 S_1}(1+\hat{K}_{C_1 S_2})\hat{R}_S(0)} \tag{22}$$

If we do not scale, together with *Equation 11*, this becomes:

$$
\begin{aligned}
S_1(0) &\gg \frac{(K_{C_1 S_2}/C_1^*)(r_{10}-r_{20})}{(\beta_{C_1 S_1}/C_1^*)(1+K_{C_1 S_2}/C_1^*)R_S(0)/R_S^*} \\
&= \frac{K_{C_1 S_2}(r_{10}-r_{20})}{R_S(0)\alpha_{C_1 S_2}}
\end{aligned}
\tag{23}
$$

In this case, the community dynamics trajectory before getting close to the $f$-zero-isocline can be approximated as a straight line (the orange dotted line) and the change in $\hat{R}_S$ can be approximated by the green segment in the inset of *Figure 3—figure supplement 2B*. Since the green segment, the orange dotted line and the red dashed line form a right angle triangle, the length of green segment can be calculated once we find the length of the red dashed line $\Delta\hat{C}_1$, which is the horizontal distance between $(\hat{C}_1(0), \hat{R}_S(0))$ and the $f$-zero-isocline and can be calculated from *Equation 16*:

$$1 - \frac{\Delta\hat{C}_1(1+\hat{K}_{C_1 S_2})}{\Delta\hat{C}_1+\hat{K}_{C_1 S_2}}\hat{R}_S(0) = 0$$

which yields

$$\Delta\hat{C}_1 = \frac{\hat{K}_{C_1 S_2}}{\hat{R}_S(0)(1+\hat{K}_{C_1 S_2})-1} \approx \frac{\hat{K}_{C_1 S_2}}{\hat{R}_S(0)(1+\hat{K}_{C_1 S_2})} \tag{24}$$

The green segment $\Delta\hat{R}_S$ is then the length of red dashed line ($\Delta\hat{C}_1$, *Equation 24* ) multiplied with $\left.\frac{d\hat{R}_S}{d\hat{C}_1}\right|_{\hat{C}_1(0)=0}$ (*Equation 20*), or

$$\Delta\hat{R}_S = \frac{(r_{20} - r_{10})\hat{K}_{C_1 S_2}}{\hat{\beta}_{C_1 S_1} S_1(0)\left(1 + \hat{K}_{C_1 S_2}\right)}$$ (25)

Note that if *Equation 22* is satisfied, $|\Delta\hat{R}_S| \ll \hat{R}_S(0)$. What is the time scale $t_f$ for the community to traverse the orange dotted line to be close to the $f$-zero-isocline? Since from *Equation 14* $\frac{d\hat{R}_S}{dt} = \left(r_{20} + r_{S_2 C_1}\frac{\hat{C}_1}{\hat{C}_1 + \hat{K}_{S_2 C_1}} - r_{10}\right)\hat{R}_S \approx (r_{20} - r_{10})\hat{R}_S$. In *Equation 14*, the second term in the parenthese can be dropped if

$$\left|\frac{r_{S_2 C_1}}{r_{20} - r_{10}}\frac{\hat{C}_1}{\hat{C}_1 + \hat{K}_{S_2 C_1}}\right| \ll 1.$$

In case II-1, before the system reaches the $f$-zero-isocline, from *Equation 24*, $\hat{C}_1 \leq \Delta\hat{C}_1 < 1/\hat{R}_S(0)$ thus

$$\left|\frac{r_{S_2 C_1}}{r_{20} - r_{10}}\frac{\hat{C}_1}{\hat{C}_1 + \hat{K}_{S_2 C_1}}\right| < \left|\frac{r_{S_2 C_1}}{r_{20} - r_{10}}\frac{\Delta\hat{C}_1}{\Delta\hat{C}_1 + \hat{K}_{S_2 C_1}}\right| < \left|\frac{r_{S_2 C_1}}{r_{20} - r_{10}}\frac{1}{1 + \hat{R}_S(0)\hat{K}_{S_2 C_1}}\right|.$$

From the top portion of *Equation 11*,

$$\frac{r_{S_2 C_1}}{r_{10} - r_{20}} = \hat{K}_{S_2 C_1} + 1$$

thus

$$\left|\frac{r_{S_2 C_1}}{r_{20} - r_{10}}\frac{\hat{C}_1}{\hat{C}_1 + \hat{K}_{S_2 C_1}}\right| < \left(\hat{K}_{S_2 C_1} + 1\right)\frac{1}{1 + \hat{R}_S(0)\hat{K}_{S_2 C_1}}.$$

According to the condition, $\hat{R}_S(0) \gg \max(1, \hat{K}_{S_2 C_1}^{-1})$. If $\hat{K}_{S_2 C_1}^{-1} > 1$, then $\hat{R}_S(0) \gg \hat{K}_{S_2 C_1}^{-1}$, $\hat{R}_S(0)\hat{K}_{S_2 C_1} \gg 1$ and $\hat{K}_{S_2 C_1} < 1$.

$$\begin{aligned}\left|\frac{r_{S_2 C_1}}{r_{20} - r_{10}}\frac{\hat{C}_1}{\hat{C}_1 + \hat{K}_{S_2 C_1}}\right| &< \left(\hat{K}_{S_2 C_1} + 1\right)\frac{1}{1 + \hat{R}_S(0)\hat{K}_{S_2 C_1}} \\ &< \frac{2}{1 + \hat{R}_S(0)\hat{K}_{S_2 C_1}} \\ &\ll 1\end{aligned}$$

If $\hat{K}_{S_2 C_1}^{-1} < 1$, then $\hat{R}_S(0) \gg 1$

$$\begin{aligned}\left|\frac{r_{S_2 C_1}}{r_{20} - r_{10}}\frac{\hat{C}_1}{\hat{C}_1 + \hat{K}_{S_2 C_1}}\right| &< \left(\hat{K}_{S_2 C_1} + 1\right)\frac{1}{1 + \hat{R}_S(0)\hat{K}_{S_2 C_1}} \\ &= \left(1 + \hat{K}_{S_2 C_1}^{-1}\right)\frac{1}{\hat{K}_{S_2 C_1}^{-1} + \hat{R}_S(0)} \\ &< \frac{2}{\hat{R}_S(0)} \\ &\ll 1\end{aligned}$$

Thus, the above approximation of *Equation 14* is valid, and we obtain $t_f \approx \ln\left(\frac{\hat{R}_S(0) + \Delta\hat{R}_S}{\hat{R}_S(0)}\right)\Big/(r_{20} - r_{10})$.

Since here $\Delta\hat{R}_S \ll \hat{R}_S(0)$ and $\ln(1 + x) \sim x$ for small $x$, together with *Equation 25*, we have

$$t_f \approx \frac{\hat{K}_{C_1 S_2}}{\hat{\beta}_{C_1 S_1} S_1(0)\left(1 + \hat{K}_{C_1 S_2}\right)\hat{R}_S(0)}$$ (26)

If unscaled, using *Equation 11*, this becomes

$$t_f \approx \frac{K_{C_1 S_2}/C_1^*}{\beta_{C_1 S_1}/C_1^*\left(1 + K_{C_1 S_2}/C_1^*\right)S_1(0)R_S(0)/R_S^*} = \frac{K_{C_1 S_2}}{\alpha_{C_1 S_2} S_2(0)}$$ (27)

## Case II-2. $\hat{R}_S(t=0)$ is comparable to 1

That is, $R_S(t=0) \approx R_S^*$. If $S_{10}$ is low, a typical example is shown in **Figure 3—figure supplement 2D**. Here, because it takes a while for $C_1$ to accumulate, during this lagging phase $\hat{R}_S(t) \approx \hat{R}_S(0) \exp(-|r_{10} - r_{20}|t)$ and there is a sharp plunge in $\hat{R}_S$ before the trajectory levels off and climbs up. Although the trajectory eventually hits the $f$-zero-isocline where the alternative pairwise model can be derived, estimating $t_f$ is more complicated. Here we consider a simpler case where $S_{10}$ is large enough so that the trajectory levels off immediately after $t = 0$, and $\hat{R}_S \approx 1$ before the trajectory hits the $f$-zero-isocline (**Figure 3—figure supplement 2A**). Since $\hat{R}_S$ decreases until $\hat{C}_1 = 1$ and from **Equation 20**, and similar to the reasoning in Case II-1, if

$$|\Delta\hat{R}_S| = \left|\frac{d\hat{R}_S}{d\hat{C}_1}\Big|_{\hat{C}_1(0)=0}\right| \times 1 = \left|\frac{(r_{20} - r_{10})\hat{R}_S(0)}{\hat{\beta}_{C_1 S_1} S_1(0)}\right| \ll \hat{R}_S(0)$$

or if

$$S_1(0) \gg \frac{(r_{10} - r_{20})}{\hat{\beta}_{C_1 S_1}} \tag{28}$$

a typical trajectory moves toward the $f$-zero-isocline almost horizontally (**Figure 3—figure supplement 2A**). The unscaled form of **Equation 28** is

$$S_1(0) \gg \frac{(r_{10} - r_{20})}{\hat{\beta}_{C_1 S_1}} = \frac{(r_{10} - r_{20})C_1^*}{\beta_{C_1 S_1}} = \frac{(r_{10} - r_{20})^2 K_{S_2 C_1}}{\beta_{C_1 S_1}(r_{20} + r_{S_2 C_1} - r_{10})} \tag{29}$$

To calculate the time it takes for the trajectory to reach the $f$-zero-isocline, let $\Delta_s\hat{C}_1 = \hat{C}_1 - 1$ and $\Delta_s\hat{R}_S = \hat{R}_S - 1$ at any time point $t$ respectively represent deviation of $(\hat{C}_1(t), \hat{R}_S(t))$ away from their steady state values of $(1, 1)$. We can thus linearize **Equation 14 and Equation 15** around the steady state. Note that since at the steady state $f = 0$, thus $\Delta_s f = f$.

Rewrite **Equation 14** as

$$\frac{d\hat{R}_S}{dt} = \left(r_{20} + r_{S_2 C_1}\frac{\hat{C}_1}{\hat{C}_1 + \hat{K}_{S_2 C_1}} - r_{10}\right)\hat{R}_S = h(\hat{C}_1, \hat{R}_S).$$

We linearize this equation around the steady state $\hat{C}_1 = 1, \hat{R}_S = 1$

$$\frac{d(1+\Delta_s\hat{R}_S)}{dt} = h(1 + \Delta_s\hat{C}_1, 1 + \Delta_s\hat{R}_S) \approx h(1,1) + \Delta_s\hat{C}_1\frac{\partial h}{\partial\hat{C}_1}\Big|_{(\hat{C}_1=1,\hat{R}_S=1)} + \Delta_s\hat{R}_S\frac{\partial h}{\partial\hat{R}_S}\Big|_{(\hat{C}_1=1,\hat{R}_S=1)}.$$

At steady state, $\frac{d\hat{R}_S}{dt} = h(1,1) = 0$. Thus, $r_{20} + \frac{r_{S_2 C_1}}{1+\hat{K}_{S_2 C_1}} - r_{10} = 0$.

$$\frac{d\Delta_s\hat{R}_S}{dt} = \Delta_s\hat{C}_1\hat{R}_S r_{S_2 C_1}\frac{(\hat{C}_1 + \hat{K}_{S_2 C_1}) - \hat{C}_1}{(\hat{C}_1 + \hat{K}_{S_2 C_1})^2}\Big|_{(\hat{C}_1=1,\hat{R}_S=1)} + \Delta_s\hat{R}_S\left(r_{20} + \frac{r_{S_2 C_1}\hat{C}_1}{\hat{C}_1 + \hat{K}_{S_2 C_1}} - r_{10}\right)\Big|_{(\hat{C}_1=1,\hat{R}_S=1)}$$

$$= \Delta_s\hat{C}_1\frac{r_{S_2 C_1}\hat{K}_{S_2 C_1}}{(1+\hat{K}_{S_2 C_1})^2} + \Delta_s\hat{R}_S\left(r_{20} + \frac{r_{S_2 C_1}}{1+\hat{K}_{S_2 C_1}} - r_{10}\right) = \Delta_s\hat{C}_1\frac{r_{S_2 C_1}\hat{K}_{S_2 C_1}}{(1+\hat{K}_{S_2 C_1})^2}.$$

Thus,

$$\frac{d\Delta_s\hat{R}_S}{dt} = \Delta_s\hat{C}_1\frac{r_{S_2 C_1}\hat{K}_{S_2 C_1}}{(1 + \hat{K}_{S_2 C_1})^2} \tag{30}$$

Recall **Equations 15 and 17** as

$$\frac{d\hat{C}_1}{dt} = \hat{\beta}_{C_1 S_1}\left(1 - \frac{\hat{C}_1(1+\hat{K}_{C_1 S_2})}{\hat{C}_1 + \hat{K}_{C_1 S_2}}\hat{R}_S\right)S_1 = \hat{\beta}_{C_1 S_1}f(\hat{C}_1, \hat{R}_S)S_1.$$

Linearize around the steady state $\hat{C}_1 = 1, \hat{R}_S = 1$ (note $f(1,1)=0$):

$$\frac{d(1+\Delta_s \hat{C}_1)}{dt} = \hat{\beta}_{C_1 S_1} S_1 \left( \Delta_s \hat{C}_1 \frac{\partial f}{\partial \hat{C}_1} \Big|_{(\hat{C}_1=1, \hat{R}_S=1)} + \Delta_s \hat{R}_S \frac{\partial f}{\partial \hat{R}_S} \Big|_{(\hat{C}_1=1, \hat{R}_S=1)} \right)$$

$$= -\hat{\beta}_{C_1 S_1} S_1 \left( \Delta_s \hat{C}_1 \left( \frac{(1+\hat{K}_{C_1 S_2})(\hat{C}_1+\hat{K}_{C_1 S_2}) - \hat{C}_1(1+\hat{K}_{C_1 S_2})}{(\hat{C}_1+\hat{K}_{C_1 S_2})^2} \hat{R}_S \right) \Big|_{(\hat{C}_1=1, \hat{R}_S=1)} + \Delta_s \hat{R}_S \frac{\hat{C}_1(1+\hat{K}_{C_1 S_2})}{\hat{C}_1+\hat{K}_{C_1 S_2}} \Big|_{(\hat{C}_1=1, \hat{R}_S=1)} \right)$$

$$= -\hat{\beta}_{C_1 S_1} S_1 \left( \Delta_s \hat{C}_1 \frac{\hat{K}_{C_1 S_2}}{1+\hat{K}_{C_1 S_2}} + \Delta_s \hat{R}_S \right).$$

Thus,

$$\frac{d\Delta_s \hat{C}_1}{dt} = -\hat{\beta}_{C_1 S_1} \left( \Delta_s \hat{C}_1 \frac{\hat{K}_{C_1 S_2}}{1+\hat{K}_{C_1 S_2}} + \Delta_s \hat{R}_S \right) S_1 \tag{31}$$

Similar to the above calculation, we expand $f$ (**Equation 16**) around steady state 0,

$$\Delta_s f = f - 0 = \Delta_s \hat{C}_1 \frac{\partial f}{\partial \hat{C}_1} \Big|_{(\hat{C}_1=1, \hat{R}_S=1)} + \Delta_s \hat{R}_S \frac{\partial f}{\partial \hat{R}_S} \Big|_{(\hat{C}_1=1, \hat{R}_S=1)}$$

$$= -\Delta_s \hat{C}_1 \frac{\hat{K}_{C_1 S_2}}{1+\hat{K}_{C_1 S_2}} - \Delta_s \hat{R}_S \tag{32}$$

Utilizing **Equation 30**, **Equation 31**, and **Equation 32**,

$$\frac{d\Delta_s f}{dt} = \frac{df}{dt} = \frac{-d}{dt} \left( \Delta_s \hat{C}_1 \frac{\hat{K}_{C_1 S_2}}{1+\hat{K}_{C_1 S_2}} + \Delta_s \hat{R}_S \right)$$

$$= \frac{\hat{K}_{C_1 S_2} \hat{\beta}_{C_1 S_1}}{1+\hat{K}_{C_1 S_2}} \left( \Delta_s \hat{C}_1 \frac{\hat{K}_{C_1 S_2}}{1+\hat{K}_{C_1 S_2}} + \Delta_s \hat{R}_S \right) S_1 - \Delta_s \hat{C}_1 \frac{r_{S_2 C_1} \hat{K}_{S_2 C_1}}{(1+\hat{K}_{S_2 C_1})^2} \tag{33}$$

$$= -\frac{\hat{K}_{C_1 S_2} \hat{\beta}_{C_1 S_1} S_1}{1+\hat{K}_{C_1 S_2}} f - \Delta_s \hat{C}_1 \frac{r_{S_2 C_1} \hat{K}_{S_2 C_1}}{(1+\hat{K}_{S_2 C_1})^2}$$

Taking the derivative of both sides, and using **Equation 31** and **Equation 32**, we have

$$\frac{d^2 f}{dt^2} = -\frac{\hat{K}_{C_1 S_2} \hat{\beta}_{C_1 S_1}}{1+\hat{K}_{C_1 S_2}} \frac{d(S_1 f)}{dt} + \frac{r_{S_2 C_1} \hat{K}_{S_2 C_1}}{(1+\hat{K}_{S_2 C_1})^2} \hat{\beta}_{C_1 S_1} \left( \Delta_s \hat{C}_1 \frac{\hat{K}_{C_1 S_2}}{1+\hat{K}_{C_1 S_2}} + \Delta_s \hat{R}_S \right) S_1$$

$$= -\frac{\hat{K}_{C_1 S_2} \hat{\beta}_{C_1 S_1}}{1+\hat{K}_{C_1 S_2}} \frac{d(S_1 f)}{dt} - \frac{\hat{\beta}_{C_1 S_1} r_{S_2 C_1} \hat{K}_{S_2 C_1}}{(1+\hat{K}_{S_2 C_1})^2} f S_1$$

The solution to the above equation is:

$$f = \exp\left( -\frac{b}{2r_{10}} e^{r_{10} t} - \frac{r_{10} t}{2} \right) \cdot \left( D_1 M\left( \frac{1}{2} + \frac{a}{br_{10}}, 0, \frac{e^{r_{10} t} b}{r_{10}} \right) + D_2 W\left( \frac{1}{2} + \frac{a}{br_{10}}, 0, \frac{e^{r_{10} t} b}{r_{10}} \right) \right)$$

where $a = r_{S_2 C_1} \hat{K}_{S_2 C_1} \hat{\beta}_{C_1 S_1} S_{10} / (1+\hat{K}_{S_2 C_1})^2$ and $b = \hat{\beta}_{C_1 S_1} \hat{K}_{C_1 S_2} S_{10} / (1+\hat{K}_{C_1 S_2})$ are two positive constants. $D_1$ and $D_2$ are two constants that can be determined from the initial conditions of $\hat{R}_S$ and $\hat{C}_1$. $M(\kappa, \mu, z)$ and $W(\kappa, \mu, z)$ are Whittaker functions with argument **z**. As $z \to \infty$ (http://dlmf.nist.gov/13.14.E20 and http://dlmf.nist.gov/13.14.E21)

$M(\kappa, \mu, z) \sim \exp(z/2) z^{-\kappa}$.
$W(\kappa, \mu, z) \sim \exp(-z/2) z^{\kappa}$.
Thus when $e^{r_{10} t} b / r_{10} \gg 1$,

$$f \sim D_1 \exp\left[ -\frac{b}{2r_{10}} e^{r_{10} t} - \frac{r_{10} t}{2} + \frac{e^{r_{10} t} b}{2r_{10}} \right] \left( \frac{e^{r_{10} t} b}{r_{10}} \right)^{-\left( \frac{1}{2} + \frac{a}{br_{10}} \right)} + D_2 \exp\left[ -\frac{b}{2r_{10}} e^{r_{10} t} - \frac{r_{10} t}{2} - \frac{e^{r_{10} t} b}{2r_{10}} \right] \left( \frac{e^{r_{10} t} b}{r_{10}} \right)^{\left( \frac{1}{2} + \frac{a}{br_{10}} \right)}$$

$$= \left( \frac{b}{r_{10}} \right)^{-\left( \frac{1}{2} + \frac{a}{br_{10}} \right)} D_1 \exp\left( -\left( r_{10} + \frac{a}{b} \right) t \right) + \left( \frac{b}{r_{10}} \right)^{\left( \frac{1}{2} + \frac{a}{br_{10}} \right)} D_2 \exp\left( \frac{-b}{r_{10}} e^{r_{10} t} + \frac{a}{b} t \right)$$

The second term approaches zero much faster compared to the first term due to the negative exponent with an exponential term. Thus,

$$f \propto \exp\left( -\left( r_{10} + \frac{a}{b} \right) t \right) = \exp\left( -\left( r_{10} + \frac{r_{S_2 C_1} \hat{K}_{S_2 C_1} \hat{\beta}_{C_1 S_1} S_{10} / (1+\hat{K}_{S_2 C_1})^2}{\hat{\beta}_{C_1 S_1} \hat{K}_{C_1 S_2} S_{10} / (1+\hat{K}_{C_1 S_2})} \right) t \right)$$

$$= \exp\left( -\left( r_{10} + \frac{r_{S_2 C_1} \hat{K}_{S_2 C_1} (1+\hat{K}_{C_1 S_2})}{\hat{K}_{C_1 S_2} (1+\hat{K}_{S_2 C_1})^2} \right) t \right) \tag{34}$$

Thus, when $e^{r_{10} t} b / r_{10} \gg 1$, $\Delta_s f = f$ approaches zero at a rate of $r_{10} + \frac{r_{S_2 C_1} \hat{K}_{S_2 C_1} (1+\hat{K}_{C_1 S_2})}{\hat{K}_{C_1 S_2} (1+\hat{K}_{S_2 C_1})^2}$. Therefore, as a conservative estimation, for

$$t \gg r_{10}^{-1} \tag{35}$$

the community is sufficiently close to **f**-zero-isocline.

## Case II-3. $\hat{R}_S(t=0) \ll 1/\left(1+\hat{K}_{C_1 S_2}\right)$ or $R_S(0) \ll \beta_{C_1 S_1}/\alpha_{C_1 S_2}$

Similar to Case II-2, if *Equation 28* is satisfied, a typical trajectory is illustrated in *Figure 3—figure supplement 2C* where the trajectory decreases slightly until $\hat{C}_1 = 1$. $\hat{C}_1$ then increases to much greater than one before the system reaches the **f**-zero-isocline. $t_f$ can then be estimated from $t_{f1}$, the time it takes for $\hat{C}_1$ to reach 1 and $t_{f2}$, the time takes for $\hat{R}_S$ to increase to $1/\left(1+\hat{K}_{C_1 S_2}\right)$. Using *Equation 15*, since $\hat{R}_S$ decreases very little, and $\hat{R}_S(t=0) \ll 1/\left(1+\hat{K}_{C_1 S_2}\right)$,

$$\frac{d\hat{C}_1}{dt} \approx \hat{\beta}_{C_1 S_1} S_1 = \hat{\beta}_{C_1 S_1} S_{10} \exp(r_{10}t)$$

Therefore, $\hat{C}_1 \approx \frac{\hat{\beta}_{C_1 S_1} S_1(0)}{r_{10}}(e^{r_{10}t} - 1)$.

During $t_{f1}$, $\hat{C}_1$ increases from 0 to 1. Thus, $1 \approx \frac{\hat{\beta}_{C_1 S_1} S_1(0)}{r_{10}}(e^{r_{10}t_{f1}} - 1)$. $t_{f1} \approx \ln(r_{10}/(\hat{\beta}_{C_1 S_1} S_{10}) + 1)/r_{10}$.
If

$$S_1(0) \gg \frac{r_{10}}{\hat{\beta}_{C_1 S_1}} \tag{36}$$

$t_{f1} \approx (\hat{\beta}_{C_1 S_1} S_{10})^{-1} \ll r_{10}^{-1}$.
Using *Equation 14* and since $\hat{C}_1$ is very large,
$\frac{d\hat{R}_S}{dt} = \left(r_{20} + r_{S_2 C_1}\frac{\hat{C}_1}{\hat{C}_1 + \hat{K}_{S_2 C_1}} - r_{10}\right)\hat{R}_S \approx (r_{20} + r_{S_2 C_1} - r_{10})\hat{R}_S$.
This yields
$t_{f2} \approx \ln\left(\frac{1}{\hat{R}_S(0)\left(1+\hat{K}_{C_1 S_2}\right)}\right)/(r_{20} + r_{S_2 C_1} - r_{10})$,
and a conservative estimation of $t_f$ is

$$t_f \approx 1/r_{10} + \ln\left(\frac{1}{\hat{R}_S(0)\left(1+\hat{K}_{C_1 S_2}\right)}\right)/(r_{20} + r_{S_2 C_1} - r_{10}) \tag{37}$$

In the unscaled form, this becomes:

$$
\begin{aligned}
t_f &\approx \frac{1}{r_{10}} + \ln\left(\left(1+\frac{K_{C_1 S_2}}{C_1^*}\right)\frac{R_S(0)}{R_S^*}\right)\Big/(r_{10} - r_{20} - r_{S_2 C_1}) \\
&= \frac{1}{r_{10}} + \ln\left(\frac{\left(1+\frac{K_{C_1 S_2}}{C_1^*}\right)R_S(0)}{\frac{\beta_{C_1 S_1}}{\alpha_{C_1 S_2}}\left(1+\frac{K_{C_1 S_2}}{C_1^*}\right)}\right)\Big/(r_{10} - r_{20} - r_{S_2 C_1}) = \frac{1}{r_{10}} + \frac{\ln\left(\frac{\alpha_{C_1 S_2} R_S(0)}{\beta_{C_1 S_1}}\right)}{(r_{10} - r_{20} - r_{S_2 C_1})}
\end{aligned}
\tag{38}
$$

## Case III: $r_{10} < r_{20}$

In this case, supplier **S₁** always grows slower than **S₂**. As $t \to \infty$, $R_S = S_2/S_1 \to \infty$ and $C_1 \to 0$. The phase portrait is separated into two parts by the **f**-zero-isocline (*Figure 3—figure supplement 2E*), where, as in *Equation 12*,
$f(C_1, R_S) = 1 - \frac{\alpha_{C_1 S_2}}{\beta_{C_1 S_1}}\frac{C_1}{C_1 + K_{C_1 S_2}}R_S = 0$ or $R_S = \frac{\beta_{C_1 S_1}}{\alpha_{C_1 S_2}}\left(1+\frac{K_{C_1 S_2}}{C_1}\right)$.
Note that the asymptotic value of $R_S$ (black dotted line, *Figure 3—figure supplement 2E–H*) is

$$R_S(C_1 \to \infty) = \beta_{C_1 S_1}/\alpha_{C_1 S_2} \tag{39}$$

From *Equation 9*, $dR_S/dt > 0$. From *Equation 8*, below the $f$-zero-isocline, $dC1/dt > 0$ and above the $f$-zero-isocline, $dC1/dt < 0$. Thus if the system starts from $(0, R_S(0))$, the phase portrait dictates that it moves with a positive slope until a time of a scale $t_f$ when it hits the $f$-zero-isocline, after which it moves upward to the left closely along the $f$-zero-isocline (*Figure 3—figure supplement 2E*). After

$t_f$, the alternative pairwise model can be derived. Although $t_f$ is difficult to estimate in general, it is possible for the following cases.

## Case III-1. $R_S(0) \gg \beta_{C_1 S_1}/\alpha_{C_1 S_2}$

Similar to Case II-2, if $S_{10}$ is small, there is a lagging phase during which the trajectory rises steeply before leveling off (*Figure 3—figure supplement 2H*). Although the alternative pairwise model can be derived once the trajectory hits the $f$-zero-isocline, $t_f$ takes a complicated form. Here we consider two cases where $S_{10}$ is large enough so that we can approximate the trajectory as a straight line going through $(0, R_s(t=0))$ (*Figure 3—figure supplement 2F*). Graphically, $S_{10}$ is large enough so that the green segment in *Figure 3—figure supplement 2F*, whose length is $\Delta R_S$, is much smaller than $R_S(0)$. In other words,

$$\Delta R_S = \left. \frac{dR_S}{d(C_1/K_{C_1 S_2})} \right|_{C_1(0)=0} \times \Delta(C_1/K_{C_1 S_2}) \ll R_S(0).$$

From *Equations 8 and 9*

$$\left. \frac{dR_S}{d(C_1/K_{C_1 S_2})} \right|_{C_1(0)=0} = \left. \frac{dR_S/dt}{dC_1/dt} \right|_{t=0} K_{C_1 S_2} = \frac{(r_{20}-r_{10})R_S(0)K_{C_1 S_2}}{\beta_{C_1 S_1} S_1(0)}.$$

$\Delta(C_1/K_{C_1 S_2})$, the red segment in *Figure 3—figure supplement 2F*, is the horizontal distance between $(0, R_S(0))$ and the $f$-zero-isocline and

$$\frac{\Delta C_1}{K_{C_1 S_2}} = \frac{\beta_{C_1 S_1}}{R_S(0)\alpha_{C_1 S_2} - \beta_{C_1 S_1}}.$$

Thus, if

$$\Delta R_S = \frac{(r_{20}-r_{10})R_S(0)K_{C_1 S_2}}{\beta_{C_1 S_1} S_1(0)} \frac{\beta_{C_1 S_1}}{R_S(0)\alpha_{C_1 S_2} - \beta_{C_1 S_1}} \approx \frac{(r_{20}-r_{10})K_{C_1 S_2}}{S_1(0)\alpha_{C_1 S_2}} \ll R_S(0),$$

or

$$S_1(0) \gg \frac{(r_{20}-r_{10})K_{C_1 S_2}}{\alpha_{C_1 S_2} R_S(0)} \tag{40}$$

then from *Equation 9* and $r_{20} > r_{10}$, the upper bound of $t_f$ can be calculated as

$$\begin{aligned} t_f &\approx \ln\left(\frac{R_S(0)+\Delta R_S}{R_S(0)}\right) \Big/ (r_{20}-r_{10}) \approx \frac{\Delta R_S}{R_S(0)(r_{20}-r_{10})} \\ &\approx \frac{K_{C_1 S_2}}{R_S(0)S_1(0)\alpha_{C_1 S_2}} = \frac{K_{C_1 S_2}}{S_2(0)\alpha_{C_1 S_2}} \end{aligned} \tag{41}$$

## Case III-2. $R_S(0) \ll \beta_{C_1 S_1}/\alpha_{C_1 S_2}$

A typical example is displayed in *Figure 3—figure supplement 2G*. The trajectory moves with a small positive slope so that the intersection of the community dynamics trajectory with the $f$-zero-isocline is near the black dotted line $\beta_{C_1 S_1}/\alpha_{C_1 S_2}$ (*Equation 39*) where $C_1/K_{C_1 S_2}$ is large. The upper bound of $t_f$ can thus be estimated from *Equation 9*:

$$\frac{dR_S}{dt} = \left(r_{20} + r_{S_2 C_1} \frac{C_1/K_{S_2 C_1}}{C_1/K_{S_2 C_1} + 1} - r_{10}\right) R_S \geq (r_{20}-r_{10})R_S$$

which yields a conservative estimate of

$$t_f \approx \ln\left(\frac{\beta_{C_1 S_1}}{\alpha_{C_1 S_2} R_S(0)}\right) \Big/ (r_{20}-r_{10}) \tag{42}$$

## Conditions for the alternative pairwise model to approximate the mechanistic model

Cases II and III showed that population dynamics of the mechanistic model could be described by the alternative pairwise model. However, since the initial condition for $C_1$ cannot be specified in

pairwise model, problems could occur. To illustrate, we examine the phase portrait of the pairwise equation

$$\frac{dS_2}{dt} = r_{20}S_2 + \frac{r_{S_2C_1}S_1}{\omega S_1 + \psi S_2}S_2 \tag{13}$$

where $\omega = 1 - \frac{K_{S_2C_1}}{K_{C_1S_2}}$, $\psi = \frac{\alpha_{C_1S_2}K_{S_2C_1}}{\beta_{C_1S_1}K_{C_1S_2}}$. From *Equations 8 and 13*,

$$\frac{dR_S}{dt} = \frac{d\left(\frac{S_2}{S_1}\right)}{dt} = \frac{\left(r_{20} + \frac{r_{S_2C_1}S_1}{\omega S_1 + \psi S_2}\right)S_2S_1 - S_2r_{10}S_1}{S_1^2} = \left(r_{20} + \frac{r_{S_2C_1}}{\omega + \psi R_S} - r_{10}\right)R_S \tag{43}$$

Below, we plot *Equation 43* under different parameters (*Figure 3—figure supplement 3*) to reveal conditions for convergence between mechanistic and pairwise models.

- Case II ($r_{S_2C_1} > r_{10} - r_{20} > 0$): steady state $R_S^*$ exists for mechanistic model.

If $\omega = 1 - K_{S_2C_1}/K_{C_1S_2} \geq 0$ (*Figure 3—figure supplement 3A*): When $R_S S^*$, $dR_S/dt$ is positive. When $R_S S^*$, $dR_S/dt$ is negative. Thus, wherever the initial $R_S$, it will always converge toward the only steady state $R_S^*$ of the mechanistic model.

If $\omega < 0$ (*Figure 3—figure supplement 3B*): $\omega + \psi R_S = 0$ or $R_S = -\omega/\psi$ creates singularity. Pairwise model $R_S$ will only converge toward the mechanistic model steady state if

$$R_S(0) > -\omega/\psi \tag{44}$$

- Case III ($r_{10} < r_{20}$): $R_S$ increases exponentially in mechanistic model (*Equation 9*). Thus, $C_1$ will decline toward zero as **C₁** is consumed by **S₂** whose relative abundance over **S₁** exponentially increases. Hence, according to *Equation 9*, $R_S$ eventually increases exponentially at a rate of $r_{20} - r_{10}$.

If $\omega \geq 0$ (*Figure 3—figure supplement 3C*): *Equation 43* $\frac{dR_S}{dt} = \left(r_{20} + \frac{r_{S_2C_1}}{\omega + \psi R_S} - r_{10}\right)R_S > 0$. Thus, *Equation 43*, which is based on alternative pairwise model, also predicts that $R_S$ will eventually increase exponentially at a rate of $r_{20} - r_{10}$, similar to the mechanistic model.

If $\omega < 0$ (*Figure 3—figure supplement 3D*): $R_S(0) > -\omega/\psi$ (*Equation 44*) is required for unbounded increase in $R_S$ (similar to the mechanistic model). Otherwise, $R_S$ converges to an erroneous value instead.

## Conditions under which a saturable L-V pairwise model can represent one species influencing another via two reusable mediators

Here, we examine a simple case where **S₁** releases reusable $C_1$ and $C_2$, and $C_1$ and $C_2$ additively affect the growth of **S₂** (see example in *Figure 5*). Similar to *Figure 3A*, the mechanistic model is:

$$\begin{cases} S_1 &= S_{10}\exp(r_{10}t) \\ \frac{dS_2}{dt} &= \left(r_{20} + \frac{r_{S_2C_1}S_1}{S_1 + K_{S_2C_1}r_{10}/\beta_{C_1S_1}} + \frac{r_{S_2C_2}S_1}{S_1 + K_{S_2C_2}r_{10}/\beta_{C_2S_1}}\right)S_2 \end{cases} \tag{45}$$

Now the question is whether the saturable L-V pairwise model

$$\begin{cases} S_1 &= S_{10}\exp(r_{10}t) \\ \frac{dS_2}{dt} &= \left(r_{20} + r_{21}\frac{S_1}{S_1 + K_{21}}\right)S_2 \end{cases}$$

can be a good approximation.

For simplicity, let's define $K_{C1} = K_{S_2C_1}r_{10}/\beta_{C_1S_1}$ and $K_{C2} = K_{S_2C_2}r_{10}/\beta_{C_2S_1}$. Small $K_{Ci}$ means large potency (e.g. small $K_{C2}$ can be caused by small $K_{S_2C_2}$ which means low $C_2$ required to achieve half maximal effect on **S₂**, and/or large synthesis rate $\beta_{C_2S_1}$). Since $S_1$ from pairwise and mechanistic models are identical, we have

$$\begin{aligned}
\bar{D} &= \frac{1}{2T}\int_T \left|\log_{10}\left(S_{2,pair}\right) - \log_{10}\left(S_{2,mech}\right)\right|dt \\
&= \frac{1}{2T\ln(10)}\int_T \left|\ln\left(S_{2,pair}\right) - \ln\left(S_{2,mech}\right)\right|dt \\
&= \frac{1}{2T\ln(10)}\int_T \left|\int_t \left(r_{20} + r_{21}\frac{S_1}{S_1+K_{21}}\right)d\tau - \int_t\left(r_{20} + \frac{r_{S_2C_1}S_1}{S_1+K_{C1}} + \frac{r_{S_2C_2}S_1}{S_1+K_{C2}}\right)d\tau\right|dt \\
&= \frac{1}{2T\ln(10)}\int_T \left|\int_t \left[r_{21}\frac{S_1}{S_1+K_{21}} - \left(\frac{r_{S_2C_1}S_1}{S_1+K_{C1}} + \frac{r_{S_2C_2}S_1}{S_1+K_{C2}}\right)\right]d\tau\right|dt
\end{aligned} \tag{46}$$

$\bar{D}$ can be close to zero when (i) $K_{C1}\approx K_{C2}$ or (ii) $\frac{r_{S_2C_1}S_1}{S_1+K_{C1}}$ and $\frac{r_{S_2C_2}S_1}{S_1+K_{C2}}$ (effects of $C_1$ and $C_2$ on $S_2$) differ dramatically in magnitude. For (ii), without loss of generality, suppose that the effect of $C_2$ on $S_2$ can be neglected. This can be achieved if (iia) $r_{S_2C_2}$ is much smaller than $r_{S_2C_1}$, or (iib) $K_{C_2}$ is large compared to $S_1$.

## Competitive commensal interaction

For the community in **Figure 6A**, our mechanistic model is:

$$\begin{aligned}
\frac{dS_1}{dt} &= \left(r_{10} + r_{S_1C_1}\frac{C_1}{C_1+K_{S_1C_1}}\right)S_1 \\
\frac{dS_2}{dt} &= \left[r_{20} + r_{S_2C_{1,2}}\frac{\left(C_1/K_{S_2C_1}\right)\left(C_2/K_{S_2C_2}\right)}{C_1/K_{S_2C_1}+C_2/K_{S_2C_2}}\left(\frac{1}{C_1/K_{S_2C_1}+1} + \frac{1}{C_2/K_{S_2C_2}+1}\right)\right]S_2 \\
\frac{dC_1}{dt} &= \beta_0 - \alpha_{C_1S_1}r_{S_1C_1}\frac{C_1}{C_1+K_{S_1C_1}}S_1 - \alpha_{C_1S_2}r_{S_2C_{1,2}}\frac{\frac{C_1}{K_{S_2C_1}}\frac{C_2}{K_{S_2C_2}}}{\frac{C_1}{K_{S_2C_1}}+\frac{C_2}{K_{S_2C_2}}}\left(\frac{1}{\frac{C_1}{K_{S_2C_1}}+1} + \frac{1}{\frac{C_2}{K_{S_2C_2}}+1}\right)S_2 \\
\frac{dC_2}{dt} &= \beta_{C_2S_1}S_1 - \alpha_{C_2S_2}r_{S_2C_{1,2}}\frac{\left(C_1/K_{S_2C_1}\right)\left(C_2/K_{S_2C_2}\right)}{C_1/K_{S_2C_1}+C_2/K_{S_2C_2}}\left(\frac{1}{C_1/K_{S_2C_1}+1} + \frac{1}{C_2/K_{S_2C_2}+1}\right)S_2
\end{aligned} \tag{47}$$

Here, $S_1$ and $S_2$ are the densities of the two species; $r_{i0}$ is the basal net growth rate of $S_i$ (negative, representing death in the absence of the essential shared resource $C_1$); $C_1$ is supplied at a constant rate $\beta_0$; $\beta_{C_2}S_1$ is the production rate of $C_2$ by $S_1$; $\alpha_{C_i}S_j$ is the amount of resource $C_i$ consumed to produce a new $S_j$ cell.

The growth of $S_2$ is controlled by $C_1$ and $C_2$. When $C_1$ is limiting ($C_1/K_{S_2C_1} \ll C_2/K_{S_2C_2}$), the fitness influence of the two chemicals on $S_2$ becomes:

$$\begin{aligned}
&r_{S_2C_{1,2}}\frac{\left(C_1/K_{S_2C_1}\right)\left(C_2/K_{S_2C_2}\right)}{C_1/K_{S_2C_1}+C_2/K_{S_2C_2}}\left(\frac{1}{C_1/K_{S_2C_1}+1} + \frac{1}{C_2/K_{S_2C_2}+1}\right) \\
&\approx r_{S_2C_{1,2}}\frac{\left(C_1/K_{S_2C_1}\right)\left(C_2/K_{S_2C_2}\right)}{C_2/K_{S_2C_2}}\left(\frac{1}{C_1/K_{S_2C_1}+1}\right) = r_{S_2C_{1,2}}\frac{C_1/K_{S_2C_1}}{C_1/K_{S_2C_1}+1} = r_{S_2C_{1,2}}\frac{C_1}{C_1+K_{S_2C_1}}
\end{aligned}$$

which is the standard Monod equation. A similar argument holds for limiting $C_2$. We have intentionally chosen very large $K_{S_2}C_2$ to ensure that the fitness effect of $C_2$ on $S_2$ is linear with respect to $C_2$. This way, we minimize the number of pairwise model parameters that need to be estimated.

For our L-V pairwise model, to capture intra-species competition, we use

$$\frac{dS_i}{dt} = b_{i0}\left(1 - \frac{S_i}{\kappa_i}\right)S_i - d_iS_i$$

where non-negative $b_{i0}$ represents the maximal birth rate of $S_i$ at nearly zero population density (no competition), and non-negative $d_i$ represents the constant death rate of $S_i$. Positive $\kappa_i$ is the 'carrying capacity' imposed by the limiting resource, and is the $S_i$ at which birth rate becomes zero. This equation can be simplified to:

$$\frac{dS_i}{dt} = (b_{i0} - d_i)\left[1 - \frac{S_i}{\kappa_i(1 - d_i/b_{i0})}\right]S_i = r_{i0}\left[1 - \frac{S_i}{\Lambda_i}\right]S_i.$$

When $\Lambda_i > 0$ (i.e. when $b_{i0} > d_i$), this resembles standard L-V model traditionally used for competitive interactions (compare to **Equation 2**; **Gause, 1934**; **Thébault and Fontaine, 2010**; **Mougi and Kondoh, 2012**).

Thus, for the competitive commensal community, we have:

$$\frac{dS_1}{dt} = b_{10}\left(1 - \frac{S_1}{\Lambda_{11}} - \frac{S_2}{\Lambda_{12}}\right)S_1 - d_1 S_1$$

$$\frac{dS_2}{dt} = (b_{20} + r_{21}S_1)\left(1 - \frac{S_1}{\Lambda_{21}} - \frac{S_2}{\Lambda_{22}}\right)S_2 - d_2 S_2 \tag{48}$$

Here, birth rate of each species is reduced by competition from the two species, and $\Lambda_{ij}$ is the carrying capacity such that a single $S_i$ individual will have a zero birth rate when encountering a total of $\Lambda_{ij}$ individuals of $S_j$. For $S_2$, We used $(b_{20} + r_{21}S_1)\left(1 - \frac{S_1}{\Lambda_{21}} - \frac{S_2}{\Lambda_{22}}\right)S_2$ instead of $b_{20}\left(1 - \frac{S_1}{\Lambda_{21}} - \frac{S_2}{\Lambda_{22}}\right)S_2 + r_{21}S_1 S_2$ so that when the shared resource is exhausted (i.e. $1 - \frac{S_1}{\Lambda_{21}} - \frac{S_2}{\Lambda_{22}} = 0$), $S_2$ does not keep growing due to the presence of $S_1$.

# Additional information

### Competing interests
WS: Reviewing editor, *eLife*. The other authors declare that no competing interests exist.

### Funding

| Funder | Author |
| --- | --- |
| Boston College | Babak Momeni |
| NIH Office of the Director | Babak Momeni<br>Wenying Shou |
| W. M. Keck Foundation | Babak Momeni<br>Li Xie<br>Wenying Shou |
| Fred Hutchinson Cancer Research Center | Li Xie<br>Wenying Shou |

The funders had no role in study design, data collection and interpretation, or the decision to submit the work for publication.

### Author contributions
BM, Conceptualization, Resources, Software, Formal analysis, Funding acquisition, Validation, Investigation, Visualization, Methodology, Writing—original draft, Project administration, Writing—review and editing; LX, Software, Formal analysis, Validation, Investigation, Visualization, Methodology, Writing—review and editing; WS, Conceptualization, Resources, Formal analysis, Supervision, Funding acquisition, Validation, Investigation, Visualization, Methodology, Writing—original draft, Project administration, Writing—review and editing

### Author ORCIDs
Babak Momeni, http://orcid.org/0000-0003-1271-5196
Li Xie, http://orcid.org/0000-0003-3397-2407
Wenying Shou, http://orcid.org/0000-0001-5693-381X

# Additional files

### Supplementary files
• Source code 1. Calculates the cost function for monocultures (i.e. the difference between the target mechanistic model dynamics and the dynamics obtained from the pairwise model).

• Source code 2. Calculates the cost function for communities (i.e. the difference between the target mechanistic model dynamics and the dynamics obtained from the saturable L-V pairwise model).

• Source code 3. Calculates the cost function for communities (i.e. the difference between the target mechanistic model dynamics and the dynamics obtained from the alternative pairwise model).

• Source code 4. Returns growth dynamics for monocultures, based on the mechanistic model.

• Source code 5. Returns growth dynamics for communities of multiple species, based on the mechanistic model.

• Source code 6. Returns growth dynamics for communities of multiple species, based on the saturable L-V pairwise model.

• Source code 7. Returns growth dynamics for communities of multiple species, based on the alternative pairwise model.

• Source code 8. Estimates monoculture parameters of pairwise model (Step 1).

• Source code 9. Estimates saturable L-V pairwise model interaction parameters (Step 2).

• Source code 10. Estimates alternative pairwise model interaction parameters (Step 2).

• Source code 11. Estimates saturable L-V pairwise model interaction parameters ($r_{21}$ and $K_{21}$) in cases where we know that $S_2$ is only affected by $S_1$, to accelerate optimization.

• Source code 12. Estimates alternative pairwise model interaction parameter ($r_{21}$) in cases where we know that $S_2$ is only affected by $S_1$ and that $K_{S2C1}=K_{C1S2}$ to accelerate optimization.

• Source code 13. Returns growth dynamics for communities of two species competing for an environmental resource while engaging in an additional interaction, based on the logistic L-V pairwise model (*Figure 6*).

• Source code 14. Estimates logistic L-V pairwise model interaction parameters for communities of two species competing for an environmental resource while engaging in an additional interaction, and compares community dynamics from pairwise and mechanistic models (*Figure 6*).

• Source code 15. Defines differential equations when using Matlab's ODE23 solver to calculate community dynamics.

• Source code 16. Example of using Matlab ODE23 solver for calculating community dynamics.

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
