## [Decision Letter]

[Editors’ note: a previous version of this study was rejected after peer review, but the authors submitted for reconsideration. The first decision letter after peer review is shown below.]

Thank you for submitting your work entitled "The validity of pairwise models in predicting community dynamics" for consideration by *eLife*. Your article has been reviewed by two peer reviewers, and the evaluation has been overseen by a Reviewing Editor, Bruce R. Levin, and a Senior Editor. The reviewers have opted to remain anonymous.

Our decision has been reached after consultation between the reviewers. Based on these discussions and the individual reviews below, we regret to inform you that your work will not be considered further for publication in *eLife*.

This review and editorial decision, has two elements. Part 1 has been put together by the Reviewing editor but is the product of a discussion between the editor and the reviewers. Part 2 are the separate comments and suggestions of the reviewers. There is a fair amount of overlap. BRL agrees with all of the critical comments and suggestions of the reviewers. Following our discussion, BRL sent the first part of this recommendation to the reviewers. Save for a typo, they approved of this collective review and recommendation.

Part 1.

Recommendation:

The subject of this report is right on. In addition to the considerable industrial and ecological and medical interest in bacterial communities, thanks to the "microbiome mania" the need to understand the processes that determine and maintain the structure of these communities has become increasingly important. Central to acquiring this understanding will doubtless be mathematical and/or computer simulations models of the population and evolutionary dynamics of bacteria and the interactions between these populations and the physical, chemical and biotic factors that determine their densities and relative frequencies in communities. How to construct and analyze the properties of these models is not at all clear at this time; the long-standing fissure between population and community ecology has yet to be breached. This report considers the problems of modeling bacterial populations and communities.

As trendy and important as this subject is, we don't consider this report to be of sufficient general interest to be published in *ELife*. Although it may be of interest to some theoreticians, we believe it would be of little interest and utility to population and evolutionary biologists and biometricians doing experimental or other empirical research with bacteria and bacterial communities. It is too abstract and doesn't address specific questions. Some readers may also see it as a complex, equation illustrated rant telling them what they already know about the limitations of pairwise models. We believe it would be more suitable for a fine but more specialized journal like PLoS Computational Biology, where the readers will be more attune to mathematical modeling of the sort considered in this report.

Collective comments and suggestions:

1) – We agree with the George Box adage, "All models are wrong, some are useful". However, we don't see how their pairwise models would be useful for understanding the processes that determine "distribution and abundance" of bacterial strains and species in communities. To be sure, even without understanding the mechanisms responsible, it would be useful to be able to predict the distribution and abundance of species from empirical estimates of the parameter of these "pairwise" models. But as we see this report as a cautionary tale saying that it is unlikely that pairwise models will be able to achieve these ends much beyond three species and if that. Moreover, even if it were possible to provide predictive pairwise models for specific simple communities, perhaps the macrobiotic of yogurt, there would be little or no generality, like to cheese or bread starters, much less natural communities.

2) Arguably (not to BRL) mechanistic models are most useful when they are wrong, when there are substantial qualitative rather than just small quantitative differences between the predictions of the model and observations in the empirical studies. That way one knows that there are fundamental errors in the biological, chemical and environmental assumptions upon which the model was based. In the best of cases, a wrong mechanistic model will point to the biological and other assumptions that have to be modified to obtain a better fit and thereby increase our understanding of the mechanisms responsible for the phenomenon under study. It is not clear how the pairwise models of the sort they are considering could achieve this end when they fit or don't fit.

3) What one means by a "mechanistic model" and mechanisms in general is in the eyes, mind and perspective of the beholder. The authors can do a better job of providing the readers with a clear distinction between what they mean by pairwise rather than mechanistic models. They present the logistic model and its extensions, like the Gause competition models, as pairwise rather than mechanistic models. We agree, but some may not. These models make the reasonable assumption that the rate of growth of populations decline as they approach the point of saturation of the environment, the parameter K. But the authors and most others interpret that parameter to reflect resource limitation, which seems mechanistic albeit not explicit in the model. Moreover, the ecological nature of the equilibrium in Logistic models and its extensions is not defined. Is that equilibrium, K, the density at stationary phase in batch culture? Or is the population at equilibrium in an environment in which resources are continually made available and bacteria and wastes are continually being removed, like a chemostat or turbidostat?

4) There is a real need to develop models that deal with the inconvenient reality of the physical structure of real habitats of bacteria. The ODE mechanistic and pairwise models considered in this report do not address this reality, which doubtless contributes to the distribution and abundance of species and strains of bacteria in natural communities.

Part 2.

*Reviewer #1:*

In this work, the authors study the validity of utilizing phenomenological models of pairwise interactions to describe the dynamics of ecological communities. To do so, they employ theory and simulations to compare the dynamics of detailed mechanistic models to the ones of appropriately parameterized pairwise models.

Improving our understanding of the capabilities and limitations of pairwise models is an important and timely goal, especially given their recent popularity in modeling microbial communities. I also appreciate the authors' approach of trying to delineate the types of mechanistic situations in which the pairwise approximation is valid.

However, the aspects of the problem that the authors focused on are not always the ones I feel are the most interesting and useful. Specifically, the authors focused on identifying situations in which a specific pairwise model can provide an exact description of the community dynamics. It is not surprising that phenomenological pairwise models often fail to capture exactly the dynamics of more complex mechanistic models. Nonetheless, there may be many situations where they are still useful, either by providing approximate description of the dynamics, or by capturing important qualitative features, such as the existence of oscillations, the set of coexisting species, presence of alternative stable states, etc. To me, understanding the conditions under which pairwise models are not even approximately correct, or make qualitatively wrong prediction is one of the important outstanding challenges in community modeling. I would encourage the authors to use their approach to provide insight into these questions, but, at the very least, they should clarify the difference between different failures of pairwise models. Additionally, they should be more explicit about cases in which no pairwise model would capture the community dynamics (e.g. in cases of interaction modification, or higher order interaction), versus ones in which the pairwise model that was considered wasn't adequate, but there may be a different one that is.

The authors also do not consider cases where interactions are mediated by externally supplied, abiotic mediators, rather than ones produced by the species themselves. The author's model is an extension of MacArthur's competition model, whose link to pairwise models have been extensively studied (e.g. Chesson, P. "MacArthur's Consumer Resource Model." Theoretical Population Biology 37, no. 1 (1990): 2638.).

What is not captured is competition of abiotic resources, such as the models common in David Tilman's work. The authors should make these distinctions clear, and put their work in the context of previous work. I also believe that extending the work to include competition for abiotic resources would add significant value to the work.

*Reviewer #2:*

The work of Momeni et al. "The validity of pairwise models in predicting community dynamics" explores conditions under which mechanistic models of species interacting via chemical mediators can be reduced to pairwise models. Pairwise models do not require a full mechanistic understanding of the nature of interactions within the community and thus use fewer parameters, which is why they are often used. The authors report that in many cases, pairwise models fail to predict, quantitatively or qualitatively, the dynamics of many species communities, which is why they should be used with caution. The authors do a good job in exploring the various scenarios under which pairwise models are not sufficient to capture these dynamics.

A point that deserves more attention however is how these results are immediately relevant for past and future studies of microbial communities? The authors provide references to several studies that employed pairwise models to simulate community dynamics. However, many of those do not model chemically mediated interactions, but rather direct interactions, such as the predator prey examples mentioned in the introduction and are thus not directly relevant to what is studied here. Can the authors provide more specific examples of works that employ pairwise models to model chemically mediated interactions?

The authors clearly describe several regimes under which pairwise models fail. However, to identify whether the community being modeled falls into these regimes, one often needs to have a quite detailed mechanistic understanding of the interactions within the community in the first place. What would then be the advantages of using a pairwise model, given that this information is available? Also, it is not clear if this is an exhaustive list of regimes that they look at or are there other regimes out there to be explored? Thus, we are left with a conundrum that the work does not address: why the results are useful for systems where we do not know all the details of the interactions?

In general, for dynamical systems it is not at all surprising that 'the devil is in the details' of interactions for many a system, as weak couplings can often have important roles. Especially for systems that are under evolutionary pressure I can see how small couplings can be essential. Thus, models with underdetermined number of parameters will often fail, especially as long as the main state variables of a system are unknown, or when these tend to shift in importance from system to system.

Thus, to me the strong 'punch' in the message of the paper is missing, especially for a wider audience outside of the ecology crowd.

As a reviewer I am part of this outside audience I am not an ecologist nor am I a theorist.

---

## [Author Response]

[Editors’ note: the author responses to the first round of peer review follow.]

*This review and editorial decision, has two elements. Part 1 has been put together by the Reviewing editor but is the product of a discussion between the editor and the reviewers. Part 2 are the separate comments and suggestions of the reviewers. There is a fair amount of overlap. BRL agrees with all of the critical comments and suggestions of the reviewers. Following our discussion, BRL sent the first part of this recommendation to the reviewers. Save for a typo, they approved of this collective review and recommendation.*

*Part 1.*

*Recommendation:*

*The subject of this report is right on. In addition to the considerable industrial and ecological and medical interest in bacterial communities, thanks to the "microbiome mania" the need to understand the processes that determine and maintain the structure of these communities has become increasingly important. Central to acquiring this understanding will doubtless be mathematical and/or computer simulations models of the population and evolutionary dynamics of bacteria and the interactions between these populations and the physical, chemical and biotic factors that determine their densities and relative frequencies in communities. How to construct and analyze the properties of these models is not at all clear at this time; the long-standing fissure between population and community ecology has yet to be breached. This report considers the problems of modeling bacterial populations and communities.*

*As trendy and important as this subject is, we don't consider this report to be of sufficient general interest to be published in eLife. Although it may be of interest to some theoreticians, we believe it would be of little interest and utility to population and evolutionary biologists and biometricians doing experimental or other empirical research with bacteria and bacterial communities.*

The famous quote of George Box (“All models are wrong, some are useful”) reflects the need to omit less important details and focus on important aspects of a system when constructing a model. This quote creates a dilemma for experimentalists: how would they decide which models are sufficiently useful? Their fears are legitimate: although all models are wrong, there are different reasons for being “wrong”. Certain models are “wrong” because details (e.g. the precise values of parameters) are off. Other models are wrong because the fundamental assumptions are wrong (e.g. blending inheritance). The first class can still yield insightful information that transcends details (e.g. the famous L-V pairwise model capturing the oscillatory dynamics of predator-prey communities). We presume that this is the reason why theoretical papers are still being written and some of them are still published in high-profile journals. Given the complexity of microbial communities, the temptation of using simple pairwise modeling by “biologists and biometricians doing experimental or other empirical research with bacteria and bacterial communities” is in fact strong (see Introduction, fourth paragraph). Thus, our work is directly relevant to people who work on microbial communities.

*It is too abstract and doesn't address specific questions. Some readers may also see it as a complex, equation illustrated rant telling them what they already know about the limitations of pairwise models. We believe it would be more suitable for a fine but more specialized journal like PLoS Computational Biology, where the readers will be more attune to mathematical modeling of the sort considered in this report.*

The specific question we have addressed is “Can we use a single equation form to represent diverse pairwise microbial interactions, as done in multispecies pairwise modeling?” The answer is “No”, as elaborated in Figure 3–Figure 6. This fundamental flaw we have uncovered is orthogonal to the known limitation of pairwise modeling (i.e. modification of pairwise interactions by a third species). To our knowledge, our findings have not been discussed in the literature. This is a major contribution, given the popularity and intellectual impact of pairwise models.

From reading reviewers’ comments, we realize that our writing may have been too abstract. Thus, we have reduced the use of equations in the main text (only keeping the essential ones) and have relegated most of our mathematical derivations to Methods so that general readers don’t have to read them, but for those who might be interested, the information is readily available. To demonstrate success or failure of pairwise modeling, we have replaced abstract figures of parameter comparisons with easier-to-read figures of population composition dynamics. Finally, since it is already known that interaction modification by a third species undermines multi-species pairwise modeling, we have cut down descriptions on this and moved it (Figure 7) from Results to Discussion.

*Collective comments and suggestions:*

*1) We agree with the George Box adage, "All models are wrong, some are useful". However, we don't see how their pairwise models would be useful for understanding the processes that determine "distribution and abundance" of bacterial strains and species in communities.*

We are arguing the opposite: pairwise models are *not* as useful as one might hope when applied to microbial communities and should be used and interpreted with great caution.

*To be sure, even without understanding the mechanisms responsible, it would be useful to be able to predict the distribution and abundance of species from empirical estimates of the parameter of these "pairwise" models. But as we see this report as a cautionary tale saying that it is unlikely that pairwise models will be able to achieve these ends much beyond three species and if that.*

As stated above, we argue that a single equation form, which is the standard of practice in pairwise modeling, often fails even for two-species microbial communities.

*Moreover, even if it were possible to provide predictive pairwise models for specific simple communities, perhaps the macrobiotic of yogurt, there would be little or no generality, like to cheese or bread starters, much less natural communities.*

The possible lack of generality among different communities does not negate the importance of being able to understand communities of interest.

*2) Arguably (not to BRL) mechanistic models are most useful when they are wrong, when there are substantial qualitative rather than just small quantitative differences between the predictions of the model and observations in the empirical studies. That way one knows that there are fundamental errors in the biological, chemical and environmental assumptions upon which the model was based. In the best of cases, a wrong mechanistic model will point to the biological and other assumptions that have to be modified to obtain a better fit and thereby increase our understanding of the mechanisms responsible for the phenomenon under study. It is not clear how the pairwise models of the sort they are considering could achieve this end when they fit or don't fit.*

We agree with you that failure of pairwise models may not be as informative as failure of mechanistic models. However, to be fair, pairwise models can be useful in some aspects. We have added this paragraph to the Discussion:

“When might pairwise modeling be useful? First, pairwise modeling has been instrumental in understanding ecological phenomena such as prey-predator oscillatory dynamics and coexistence of competing predator species (Volterra 1926; MacArthur 1970; Case and Casten 1979; Chesson 1990). […] Not surprisingly, predicting qualitative consequences of species removal or addition using pairwise modeling has encountered difficulties, especially in communities of more than three species (Mounier et al. 2008; Friedman, Higgins, and Gore 2016).”

*3) What one means by a "mechanistic model" and mechanisms in general is in the eyes, mind and perspective of the beholder. The authors can do a better job of providing the readers with a clear distinction between what they mean by pairwise rather than mechanistic models.*

This is a very good point. We have now provided a clear definition of what we mean by “mechanistic models” in the Introduction:

“We define “mechanistic models” as models that explicitly consider interaction mediators as state variables. For example, if species S1 releases a compound C1 which stimulates species S2 growth upon consumption by S2, then a mechanistic model tracks concentrations of S1, C1, and S2 (Figure 1, left panels). Note that mechanistic models used here still omit molecular details such as how chemical mediators are received and processed to affect recipient.”

*They present the logistic model and its extensions, like the Gause competition models, as pairwise rather than mechanistic models. We agree, but some may not. These models make the reasonable assumption that the rate of growth of populations decline as they approach the point of saturation of the environment, the parameter K. But the authors and most others interpret that parameter to reflect resource limitation, which seems mechanistic albeit not explicit in the model. Moreover, the ecological nature of the equilibrium in Logistic models and its extensions is not defined. Is that equilibrium, K, the density at stationary phase in batch culture? Or is the population at equilibrium in an environment in which resources are continually made available and bacteria and wastes are continually being removed, like a chemostat or turbidostat?*

In our definition, the logistic model is not considered mechanistic because it does not model interaction mediators (the limiting metabolite). We agree that the logistic model can nicely capture resource competition, as shown by Gause. To clarify (Figure 1 table), in our mechanistic model, parameter *K_SC_*is the concentration of mediator *C* where half maximal fitness effect on recipient species *S* is achieved. In our pairwise model, parameter *K_ij_*is the density of the interaction-initiating species S_j_where half maximal fitness effect on recipient species S_i_is achieved. Communities in our original manuscript were grown in a turbidostat so that no metabolites other than interaction mediators were limiting. We intentionally did so to avoid introducing additional mediators and additional parameters. However, as per reviewer 1’s request, we have added an example where two species in a batch environment compete for an abiotic metabolite while simultaneously engaging in commensalism (Figure 6). We show that although a logistic L-V pairwise model can work in some cases, it generates wrong predictions in other cases, depending on parameter choices of the mechanistic model.

*4) There is a real need to develop models that deal with the inconvenient reality of the physical structure of real habitats of bacteria. The ODE mechanistic and pairwise models considered in this report do not address this reality, which doubtless contributes to the distribution and abundance of species and strains of bacteria in natural communities.*

Spatial structure is undoubtedly important, as demonstrated in our earlier work (*eLife* 00230, *eLife* 00960). In this work, we consider a well-mixed environment. We have added a paragraph to justify our choice in Results:

“Throughout this work, we consider communities grown in a well-mixed environment where all individuals interact with each other at an equal chance. […] This in turn allows us to sometimes analytically demonstrate failures of pairwise modeling.”

If a model fails to capture microbial interactions in a well-mixed environment, a spatially-structured environment is unlikely to rescue this failure.

*Part 2.*

*Reviewer #1:*

*In this work, the authors study the validity of utilizing phenomenological models of pairwise interactions to describe the dynamics of ecological communities. To do so, they employ theory and simulations to compare the dynamics of detailed mechanistic models to the ones of appropriately parameterized pairwise models.*

*Improving our understanding of the capabilities and limitations of pairwise models is an important and timely goal, especially given their recent popularity in modeling microbial communities. I also appreciate the authors' approach of trying to delineate the types of mechanistic situations in which the pairwise approximation is valid.*

*However, the aspects of the problem that the authors focused on are not always the ones I feel are the most interesting and useful. Specifically, the authors focused on identifying situations in which a specific pairwise model can provide an exact description of the community dynamics. It is not surprising that phenomenological pairwise models often fail to capture exactly the dynamics of more complex mechanistic models.*

Our writing did at times create the impression that we were only interested in exact matches, but this is not our intention. Figure 4, Figure 5, Figure 6 and Figure 7, and Figure 3—figure supplement 4 now demonstrate qualitative failures of using a single pairwise equation form to describe diverse pairwise microbial interactions and diverse details of microbial communities. We have also revised our writing in Results to clarify our intention.

Our goal is to test whether a single form of pairwise model can qualitatively predict dynamics of species pairs engaging in various types of interactions commonly found in microbial communities (e.g. Figure 2).

We summarize our results in the Discussion:

“We first consider cases where abiotic resources are in excess. When one species affects another species via a single chemical mediator, either saturable L-V or alternative pairwise model is appropriate, depending on interaction mechanism (consumable versus reusable mediator), relative fitness of the two species, and initial conditions (Figure 3; Figure 3—figure supplement 2 to Figure 3—figure supplement 5). […] Thus, although a single equation form can work in many cases, it generates qualitatively wrong predictions in many other cases.

*Nonetheless, there may be many situations where they are still useful, either by providing approximate description of the dynamics, or by capturing important qualitative features, such as the existence of oscillations, the set of coexisting species, presence of alternative stable states, etc.*

Yes, we agree. Please see the fifth paragraph of the Discussion. We have also added the following paragraph to the Introduction:

“L-V pairwise models are popular. L-V pairwise modeling has successfully explained the oscillatory dynamics of hare and its predator lynx (Figure 1—figure supplement 1) (Volterra 1926; Wangersky 1978; “BiologyEOC – PopulationChanges” 2016). […] In the second example, if herbivores (mediators of competitive interactions between carnivores) rapidly reach steady state, herbivores can be mathematically eliminated from the mechanistic model to yield a pairwise model of competing carnivores (MacArthur 1970; Chesson 1990).

*To me, understanding the conditions under which pairwise models are not even approximately correct, or make qualitatively wrong prediction is one of the important outstanding challenges in community modeling. I would encourage the authors to use their approach to provide insight into these questions, but, at the very least, they should clarify the difference between different failures of pairwise models. Additionally, they should be more explicit about cases in which no pairwise model would capture the community dynamics (e.g. in cases of interaction modification, or higher order interaction), versus ones in which the pairwise model that was considered wasn't adequate, but there may be a different one that is.*

We have re-organized our text so that in the Discussion, we recap the two failure reasons of pairwise modeling: i) Interaction modification/higher order interactions in communities of three or more species violate the additivity assumption (this is previously known; Figure 7). In this case, even if each pairwise species interaction can be described by a pairwise model, the multispecies pairwise model will fail. ii) A single equation form cannot describe diverse pairwise interactions between microbes, thus violating the universality assumption (this is the focus of our paper; Figure 3, Figure 4, Figure 5 and Figure 6).

We would like to emphasize that we are not trying to argue that “no pairwise model would capture the community dynamics”. This is likely true for complex interactions mediated by chemicals, but a strong statement like this would require a mathematical proof, which is beyond the scope of this paper. Instead, we argue that (last paragraph of Results):

“To summarize our work, even for pairwise microbial interactions, depending on interaction mechanisms (reusable versus consumable mediator, single mediator versus multiple mediators), we will need to use a plethora of pairwise models to avoid qualitative failures in predicting which species dominate a community or whether species coexist (Figure 3, Figure 4 and Figure 5). […] Taken together, pairwise modeling is unlikely to be effective for predicting community dynamics especially if interaction mechanisms are diverse.”

*The authors also do not consider cases where interactions are mediated by externally supplied, abiotic mediators, rather than ones produced by the species themselves. The author's model is an extension of MacArthur's competition model, whose link to pairwise models have been extensively studied (e.g. Chesson, P. "MacArthur's Consumer Resource Model." Theoretical Population Biology 37, no. 1 (1990): 2638.).*

*What is not captured is competition of abiotic resources, such as the models common in David Tilman's work. The authors should make these distinctions clear, and put their work in the context of previous work. I also believe that extending the work to include competition for abiotic resources would add significant value to the work.*

Originally, we intentionally left out the influence of external nutrients (e.g. competition for abiotic resources) so that we could focus on interactions mediated by a single chemical. If failure of pairwise modeling is established in these simpler cases, then our conclusion on the limitation of pairwise modeling is already valid. Nevertheless, we have now added a section on two species competing for an abiotic resource while engaging in a commensal interaction. Again, we observed that depending on parameter choices of the mechanistic model, the best-fitting pairwise model would work in some communities, but fail in other communities (Figure 6).

*Reviewer #2:*

*The work of Momeni et al. "The validity of pairwise models in predicting community dynamics" explores conditions under which mechanistic models of species interacting via chemical mediators can be reduced to pairwise models. Pairwise models do not require a full mechanistic understanding of the nature of interactions within the community and thus use fewer parameters, which is why they are often used. The authors report that in many cases, pairwise models fail to predict, quantitatively or qualitatively, the dynamics of many species communities, which is why they should be used with caution.*

Our focus is not “many-species communities.” Instead, we show that a single pairwise equation form (as used in pairwise modeling) cannot capture diverse pairwise chemical-mediated microbial interactions. Please see the last paragraph of the Results.

*The authors do a good job in exploring the various scenarios under which pairwise models are not sufficient to capture these dynamics.*

*A point that deserves more attention however is how these results are immediately relevant for past and future studies of microbial communities?*

We have moved an originally supplementary figure to the main text. In this figure (the new Figure 2), we list many examples of microbial interactions mediated by reusable or consumable compounds, which cannot be modeled by a single form of pairwise model (Figure 3 and Figure 4 and Figure 3—figure supplement 4). Multi-mediator interactions, which pose challenges for pairwise modeling (Figure 5), are also a norm rather than an exception:

“A species often affects another species via multiple mediators (Kato et al. 2008; Yang et al. 2009; Traxler et al. 2013; Kim, Lee, and Ryu 2013). For example, a fraction of a population might die and release numerous chemicals, and some of these chemicals can simultaneously affect another individual.”

As per reviewer 1’s request, we have also added an example where two species compete for an abiotic resource while engaging in a commensal interaction. Again, we find that a pairwise model sometimes works but sometimes fails (Figure 6). Given the ubiquity of various interaction mechanisms that we consider in our work, our results are immediately relevant for studies of microbial communities.

*The authors provide references to several studies that employed pairwise models to simulate community dynamics. However, many of those do not model chemically mediated interactions, but rather direct interactions, such as the predator prey examples mentioned in the introduction and are thus not directly relevant to what is studied here.*

We have added this to our Introduction:

“Pairwise models are often used to predict how perturbations of steady-state species composition exacerbate or decline over time (May 1972; Thébault and Fontaine 2010; Mougi and Kondoh 2012; Allesina and Tang 2012; Suweis et al. 2013; Coyte, Schluter, and Foster 2015). Although most work are motivated by contact-dependent prey-predation (e.g. hare-lynx) or mutualisms (e.g. plant-pollinator) where L-V models could be identical to mechanistic models, these work do not explicitly exclude chemical-mediated interactions where species are distinct from interaction mediators. “

Relevant details of these papers are below:

(May, Nature1972)

This paper models density-dependent self-inhibition for each species (a_ii_< 0), and positive or negative inter-species interactions (a_ij_). The work is abstract, with no mentioning of particular interaction mechanisms.

(Thébault and Fontaine, Science2010)

This work examines two types of networks: trophic (all interactions are +/-) and mutualistic (all interactions are +/+). For both cases, a logistic L-V form is used for intra-species competition and a saturable L-V form is used for the effect of inter-species interactions. Aside from pest-plant and plant-pollinator images to illustrate trophic and mutualistic interactions, there is no mentioning that such model may not be applied to interactions where interaction mediators are distinct from species themselves.

(Mougi and Kondoh, Science2012)

This work argues that studies on purely mutualistic or purely trophic networks may not be adequate and examines communities with both types of interactions. The model is fairly general, with one assumption that distinguishes it from previous models: "with a biologically feasible assumption that interaction strengths decrease with increasing resource species, due to an allocation of interacting effort". This makes it closer to our alternative model (Eq. 5). The work does not mention that such models can fail for microbial communities when interactions are mediated by chemicals.

(Allesina and Tang, Nature2012)

This works uses the same framework as May, but examines what would happen when interactions are mostly trophic or mostly mutualistic (or a mix), and when more realistic network structures are imposed. They broadly speak of ecological networks, without mentioning specific mechanisms that are valid or not valid.

(Suweis et al., Nature2013)

This work examines mutualistic communities (mentioning plant-animal interactions in their motivations). This work uses two types of equations: Holling Type I (linear L-V, similar to May's equations) and Holling Type II (nonlinear, similar to (Thebault and Fontaine, 2010) or the saturable L-V). Beyond occasionally mentioning plant-animals (as an example of cooperation), they don't delve into which interaction mechanisms are valid or invalid.

(Coyte et al., Science2015)

This paper specifically mentions human gut microbiome as their motivation and as examples throughout. They use equations similar to May's for their analysis.

We do not intend to undermine these works by suggesting that they should have made this distinction clear. However, without an explicit analysis such as offered by this work, pairwise models have been applied liberally to any, including microbial, communities where interaction mediators are chemicals and not species.

*Can the authors provide more specific examples of works that employ pairwise models to model chemically mediated interactions?*

Given the universality of chemical-mediated interactions in microbial communities (Figure 2), a sizable fraction of microbial interactions are mediated by chemicals. In fact, the field of “secreted metabolomics” is devoted to identifying chemical interaction mediators. We have added this to the Introduction:

“The temptation of using pairwise models is indeed high, including in microbial communities where many interactions are mediated by chemicals (Mounier et al. 2008; Faust and Raes 2012; Stein et al. 2013; Marino et al. 2014; Coyte, Schluter, and Foster 2015).”

Coyte, K.Z., Schluter, J., and Foster, K.R. (2015). The ecology of the microbiome: Networks, competition, and stability. Science 350, 663–666.

Faust, K., and Raes, J. (2012). Microbial interactions: from networks to models. Nat. Rev. Microbiol. 10, 538–550.

Marino, S., Baxter, N.T., Huffnagle, G.B., Petrosino, J.F., and Schloss, P.D. (2014). Mathematical modeling of primary succession of murine intestinal microbiota. Proc. Natl. Acad. Sci. 111, 439–444.

Mounier, J., Monnet, C., Vallaeys, T., Arditi, R., Sarthou, A.-S., Hélias, A., and Irlinger, F. (2008). Microbial interactions within a cheese microbial community. Appl. Environ. Microbiol. 74, 172–181.

Stein, R.R., Bucci, V., Toussaint, N.C., Buffie, C.G., Rätsch, G., Pamer, E.G., Sander, C., and Xavier, J.B. (2013). Ecological Modeling from Time-Series Inference: Insight into Dynamics and Stability of Intestinal Microbiota. PLoS Comput Biol 9, e1003388.

This list is by no means exhaustive.

*The authors clearly describe several regimes under which pairwise models fail. However, to identify whether the community being modeled falls into these regimes, one often needs to have a quite detailed mechanistic understanding of the interactions within the community in the first place. What would then be the advantages of using a pairwise model, given that this information is available?*

Please see the fifth paragraph of the Discussion. Also, in Discussion:

“In summary, under certain circumstances, we may already know that microbial interaction mechanisms fall within the domain of validity for a particular pairwise model. In these cases, pairwise modeling provides the appropriate level of abstraction, and constructing a pairwise model is much easier than a mechanistic model (Figure 1).”

*Also, it is not clear if this is an exhaustive list of regimes that they look at or are there other regimes out there to be explored? Thus, we are left with a conundrum that the work does not address: why the results are useful for systems where we do not know all the details of the interactions?*

The point of our work is to caution against indiscriminative use of pairwise modeling. Our work is by no means an exhaustive list of regimes that work or not work for pairwise modeling. Nor is this our goal. By demonstrating that a single form of pairwise equation, as traditionally used in pairwise modeling, cannot qualitatively capture diverse interaction mechanisms between two microbial species, we reveal an unknown/undiscussed limitation of pairwise modeling. We hope that our work will help putting pairwise modeling to proper uses (Discussion, fifth paragraph). Even though our work emphasizes the importance of delineating interaction mechanisms, we do not argue that all details of interactions must be known for a model to be useful. In fact, a good model abstracts away nonessential details. For example, interactions mediated by a single reusable mediator can be described by a saturable L-V model without the knowledge of mediator identity. Our Figure 8 provides another example of omitting unnecessary details (Figure 8 is equivalent to Figure 8). We also added to the Introduction:

“Can a single pairwise model traditionally employed in pairwise modeling qualitatively describe diverse interactions between two microbial species?[…] On the other hand, pairwise models often fail to predict species coexistence in seven-species microbial communities (Friedman, Higgins, and Gore 2016), but this could be due to interaction modification discussed above.”

*In general, for dynamical systems it is not at all surprising that 'the devil is in the details' of interactions for many a system, as weak couplings can often have important roles. Especially for systems that are under evolutionary pressure I can see how small couplings can be essential. Thus, models with underdetermined number of parameters will often fail, especially as long as the main state variables of a system are unknown, or when these tend to shift in importance from system to system.*

*Thus, to me the strong 'punch' in the message of the paper is missing, especially for a wider audience outside of the ecology crowd.*

*As a reviewer I am part of this outside audience I am not an ecologist nor am I a theorist.*

You are right in that models with underdetermined number of parameters and state variables unsurprisingly fail (although we provide as many cases where they work as cases where they fail to work, see Figure 3, Figure 4, Figure 5, Figure 6 and Figure 7). According to this view, then no pairwise modeling paper should be published. However, the reality is that these papers are published in high-profile general-audience journals and that pairwise model remains one of the most popular tools in theoretical ecology. To our knowledge, our finding (a single pairwise model cannot describe diverse pairwise microbial interactions) has not been discussed in the literature. “One man’s trivial is another man’s fundamental”, although here we are not talking about “another man” but many other men! Our work points out a fundamental flaw in arguably one of the most influential models in ecology when applied to microbial communities, and thus deserves to be in a high-profile, general-audience journal.